# AutoQD:
# Automatic Discovery of Diverse Behaviors with Quality-Diversity Optimization

**Saeed Hedayatian[1] & Stefanos Nikolaidis[1,2]**
[1]University of Southern California
[2]Archimedes AI
{saeedhed,nikolaid}@usc.edu

## Abstract

Quality-Diversity (QD) algorithms have shown remarkable success in discovering diverse, high-performing solutions, but rely heavily on hand-crafted behavioral descriptors that constrain exploration to predefined notions of diversity. Leveraging the equivalence between policies and occupancy measures, we present a theoretically grounded approach to automatically generate behavioral descriptors by embedding the occupancy measures of policies in Markov Decision Processes. Our method, AutoQD, leverages random Fourier features to approximate the Maximum Mean Discrepancy (MMD) between policy occupancy measures, creating embeddings whose distances reflect meaningful behavioral differences. A low-dimensional projection of these embeddings that captures the most behaviorally significant dimensions can then be used as behavioral descriptors for CMA-MAE, a state of the art blackbox QD method, to discover diverse policies. We prove that our embeddings converge to true MMD distances between occupancy measures as the number of sampled trajectories and embedding dimensions increase. Through experiments in multiple continuous control tasks we demonstrate AutoQD's ability in discovering diverse policies without predefined behavioral descriptors, presenting a well-motivated alternative to prior methods in unsupervised Reinforcement Learning and QD optimization. Our approach opens new possibilities for open-ended learning and automated behavior discovery in sequential decision making settings without requiring domain-specific knowledge. Source code is available at https://github.com/conflictednerd/autoqd-code.

## 1 Introduction

Traditional optimization methods, focused solely on finding optimal solutions, often fail to capture the rich diversity of possible solutions that could be valuable in different contexts. Quality-Diversity (QD) optimization addresses this limitation by generating collections of solutions that are both high-performing and behaviorally diverse (Cully et al., 2015b; Pugh et al., 2016). This approach has demonstrated success across different domains including robot locomotion (Duarte et al., 2017; Cully et al., 2015a), game level and scenario generation (Gravina et al., 2019; Bhatt et al., 2022), protein design (Boige et al., 2023), and even image generation (Fontaine et al., 2021).

Building on these successful applications, we focus on sequential decision-making tasks where we seek diverse and high-quality policies, a setting commonly referred to as Quality-Diversity Reinforcement Learning (QD-RL) (Tjanaka et al., 2022b; Nilsson and Cully, 2021; Pierrot et al., 2022). Here, the importance of behavioral diversity stems from two key considerations. First, diverse policies provide robustness against changing conditions–when one policy fails, alternatives with different behavioral characteristics might succeed. Second, diversity is crucial for open-ended learning, where the goal extends beyond solving predefined problems to continually discovering novel capabilities and behaviors (Lehman and Stanley, 2011).

A fundamental limitation of QD algorithms, particularly challenging in such tasks, is their reliance on hand-crafted *behavior descriptors* (BDs). Behavior descriptors are functions that map policies to

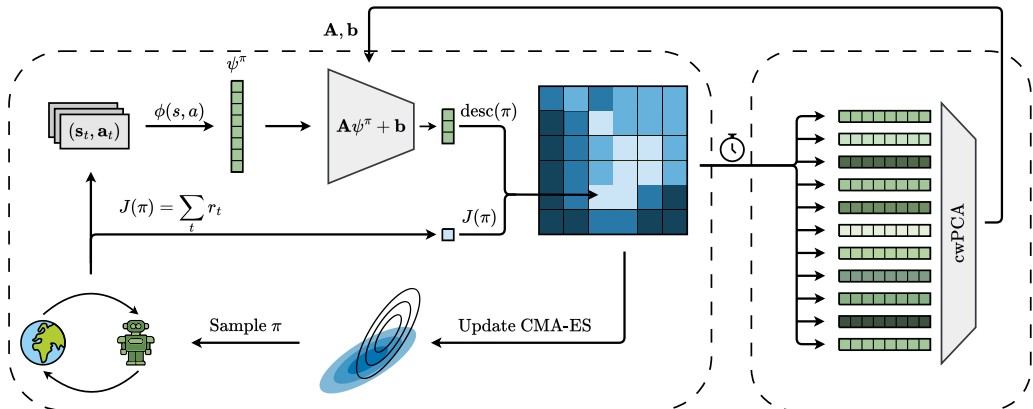

Figure 1: **Overview of AutoQD.** *Left:* Policy parameters are sampled from a CMA-ES instance and evaluated in the environment. The collected trajectories are embedded via a random Fourier features map $\phi$ to produce the policy embedding $\psi^\pi$, which is then projected to a low-dimensional descriptor using the affine map $\mathbf{A}\psi^\pi + \mathbf{b}$. The policy is added to the archive based on its return $J(\pi)$ and descriptors $\mathrm{desc}(\pi)$, and CMA-ES updates its distribution based on the improvement made to the archive. *Right:* Periodically, embeddings from the archive are used to update $\mathbf{A}$ and $\mathbf{b}$ via cwPCA.

low-dimensional vectors characterizing their behavior. For example, when designing controllers for a bipedal robot, researchers typically define BDs based on foot contact patterns, which allows them to characterize behaviors such as walking, jumping, and hopping. Hand-crafting BDs require substantial domain knowledge, which becomes increasingly difficult as task complexity grows. Furthermore, they constrain the diversity of discovered policies to variations along predefined dimensions, potentially missing interesting behavioral variations (Grillotti and Cully, 2022a).

In this paper, we present a theoretically principled approach to automatically generating behavior descriptors. Our method is based on the concept of occupancy measures, which captures the expected discounted visitation frequency of state-action pairs when following a policy. Crucially, under standard assumptions in fully-observable environments, there exists a one-to-one correspondence between policies and their occupancy measures (Puterman, 2014), making them ideal representations of behaviors as they fully characterize a policy. This differentiates our method from prior work that use human data (Ding et al., 2024) or proxy objectives such as state reconstruction (Grillotti and Cully, 2022a), to define BDs, and a wide range of other methods from the RL literature that typically use information theoretic objectives to train a fixed number of policies to be maximally different or distinguishable (Eysenbach et al., 2019; Kumar et al., 2020).

Our key insight is that by embedding occupancy measures into finite-dimensional vector spaces where distances approximate the Maximum Mean Discrepancy (MMD) between the occupancy measures, we can create behaviorally meaningful representations. These representations can then be further reduced to lower-dimensional behavior descriptors for QD optimization. Our approach, AutoQD, addresses several limitations of existing QD methods. It does not require manual specification of behavior descriptors and can potentially discover unexpected behavioral variations. Furthermore, when paired with a state-of-the-art blackbox QD algorithm, it enables us to discover thousands of policies covering a continuous behavior space.

Our main contributions are: (1) Developing a method to efficiently embed occupancy measures of policies from sampled trajectories (Sec. 3.1). (2) Formally showing how the distances between these embeddings approximate the MMD distances between occupancy measures (Theorem 1). (3) Proposing an iterative algorithm that alternates between QD optimization and behavior descriptor refinement (Sec. 3.2). (4) Demonstrating empirically that our approach discovers diverse, high-performing policies without requiring hand-crafted descriptors (Sec. 4).

## 2 BACKGROUND

### 2.1 MARKOV DECISION PROCESSES AND POLICY OPTIMIZATION

Following the established terminology in RL, we consider Markov Decision Processes (MDPs), defined by the tuple $(\mathcal{S}, \mathcal{A}, P, R, \gamma)$, where $\mathcal{S}$ is the state space, $\mathcal{A}$ is the action space, $P(\mathbf{s}'|\mathbf{s}, \mathbf{a})$ is the transition probability, $R(\mathbf{s}, \mathbf{a})$ is the reward function, and $\gamma \in (0, 1)$ is the discount factor. A policy $\pi$ is a function of the state, either deterministic ($\pi : \mathcal{S} \to \mathcal{A}$) or stochastic ($\pi : \mathcal{S} \to \Delta(\mathcal{A})$) representing an agent. The goal in RL is to find a policy that maximizes the expected discounted return $J(\pi) = \mathbb{E}_\pi[\sum_{t=0}^\infty \gamma^t R(\mathbf{s}_t, \mathbf{a}_t)]$.

A key concept in RL is the occupancy measure, which arises naturally when studying solutions to MDPs. For a policy $\pi$, its occupancy measure $\rho^\pi$ is a distribution over state-action pairs defined as:

$$\rho^\pi(\mathbf{s}, \mathbf{a}) = (1 - \gamma) \sum_{t=0}^\infty \gamma^t P(S_t = \mathbf{s}, A_t = \mathbf{a}|\pi) \tag{1}$$

where $P(S_t = \mathbf{s}, A_t = \mathbf{a}|\pi)$ is the probability of visiting state-action pair $(\mathbf{s}, \mathbf{a})$ at time $t$ when following policy $\pi$. Intuitively, $\rho^\pi(\mathbf{s}, \mathbf{a})$ represents the discounted visitation probability of $(\mathbf{s}, \mathbf{a})$ under policy $\pi$. The occupancy measure is fundamental to reinforcement learning as many quantities of interest, including the expected return $J(\pi)$, can be expressed as expectations under this measure. Importantly, under standard assumptions in fully-observable MDPs, there exists a one-to-one correspondence between Markovian policies and their occupancy measures (see Sec. 6.9.1 of Puterman (2014)), making occupancy measures a complete characterization of policy behavior.

### 2.2 QUALITY-DIVERSITY OPTIMIZATION

Quality-Diversity (QD) optimization aims to discover a collection of solutions that are both high-performing and behaviorally diverse. Unlike traditional optimization, which focuses on a single optimal solution, QD maintains an archive $\mathbb{A}$ of solutions, each associated with both a performance measure and a behavior descriptor. In QD reinforcement learning (QD-RL), a solution is the parameters of a policy, typically represented as a neural network. The performance of a policy is its expected return, $J(\pi) = \mathbb{E}_\pi[\sum_{t=0}^\infty \gamma^t R(\mathbf{s}_t, \mathbf{a}_t)]$, which we refer to as the *fitness*. A *behavior descriptor* is a function $\mathrm{desc} : \Pi \to \mathcal{B}$ that maps policies to a behavior space $\mathcal{B} \subseteq \mathbb{R}^k$. The goal of QD optimization is to find, for each behavior vector $\mathbf{b} \in \mathcal{B}$, a policy $\pi_\mathbf{b}$ that satisfies $\mathrm{desc}(\pi_\mathbf{b}) = \mathbf{b}$ and maximizes the objective among all such policies. In practice, the behavior space $\mathcal{B}$ is divided into a finite number of cells, called an *archive* $\mathbb{A}$ with the QD goal being to fill each cell with the best solution. This objective is formalized by the *QD score*, defined as $\mathrm{QDScore}(\mathbb{A}) = \sum_{\pi \in \mathbb{A}} J(\pi)$, which is the total fitness of all policies in the archive. QD algorithms employ various optimization techniques including random mutations (Cully et al., 2015b), evolutionary strategies (Fontaine et al., 2020), and gradient-based methods (Nilsson and Cully, 2021) to maximize this score.

In this work, we use CMA-MAE (Fontaine and Nikolaidis, 2023), which applies the Covariance Matrix Adaptation Evolution Strategy (CMA-ES) (Hansen, 2016) to QD optimization. CMA-MAE runs multiple CMA-ES optimizers in parallel, each maintaining a Gaussian distribution over policy parameters. In each iteration, we sample a batch of policies from the Gaussian, evaluate their fitness, and map them into the archive via their behavior descriptors. The algorithm then ranks the policies based on their improvement to the archive and uses this ranking to update the parameters of CMA-ES. This iterative update implicitly performs natural gradient ascent on (a reformulation of) the QD score (Fontaine and Nikolaidis, 2021), enabling efficient optimization of both quality and diversity.

### 2.3 MAXIMUM MEAN DISCREPANCY

To quantify the differences between policy behaviors, we turn to the *Maximum Mean Discrepancy* (MMD), a metric for comparing probability distributions. Intuitively, MMD measures the difference of two distributions by comparing statistics of their samples. Given two distributions $P$ and $Q$ over a space $\mathcal{X}$, and a feature map $\phi : \mathcal{X} \to \mathbb{R}^D$, the MMD is defined as:

$$\mathrm{MMD}(P, Q) = \|\mathbb{E}_{X \sim P}[\phi(X)] - \mathbb{E}_{Y \sim Q}[\phi(Y)]\| \tag{2}$$

When the feature map corresponds to a characteristic kernel, such as the Gaussian kernel $k(x, y) = \exp(-\|x - y\|^2/(2\sigma^2))$, MMD defines a metric over the space of probability distributions: it is non-negative, symmetric, satisfies the triangle inequality, and is zero if and only if the distributions are identical. The MMD can be computed using the "kernel trick" (Schölkopf and Smola, 2002) with a positive definite kernel $k(x, y)$, allowing for implicit feature maps even in infinite-dimensional spaces.

While there are different ways of measuring distances between distributions, we chose MMD due to its desirable properties that allow us to obtain embeddings of the distributions in a computationally efficient manner. Notably, the MMD with a Gaussian kernel can be efficiently approximated using random Fourier features (Rahimi and Recht, 2007), providing a finite-dimensional embedding that preserves the geometry of the original kernel space.

## 3 METHOD

Our method, AutoQD, automatically discovers behavior descriptors for quality-diversity optimization in sequential decision-making domains. The key insight is to use occupancy measures to characterize policy behaviors, and then extract low-dimensional BDs that capture the main variations in policy behavior. The method operates in three steps: (1) embedding policies into a space where distances approximate behavioral differences, (2) extracting low-dimensional BDs from these embeddings, and (3) using these descriptors with a standard QD algorithm (CMA-MAE) to discover diverse policies.

### 3.1 POLICY EMBEDDING VIA RANDOM FEATURES

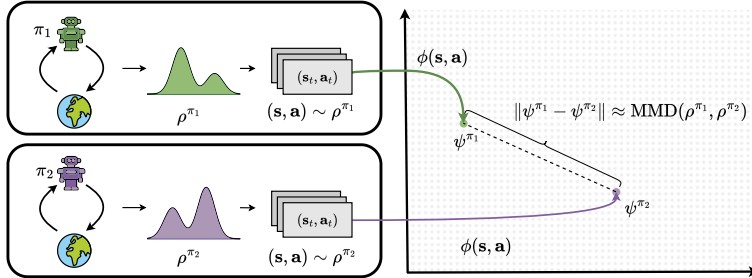

Figure 2: **Overview of the proposed policy embedding.** Each policy $\pi_i$ induces an occupancy measure $\rho^{\pi_i}$ over state-action pairs. From sampled trajectories, a feature map $\phi$ embeds the policies into a vector space. Theorem 1 guarantees that the Euclidean distance between embeddings approximates the Maximum Mean Discrepancy (MMD) between the corresponding occupancy measures.

To explore diverse behavioral variations, we embed each policy into a finite-dimensional space where Euclidean distances approximate the MMD between the occupancy measures. Since occupancy measures fully characterize policy behavior, and MMD with a Gaussian kernel defines a valid metric over them, this distance provides a meaningful measure of behavioral similarity. The challenge is that the Gaussian kernel corresponds to an infinite dimensional feature map (Schölkopf and Smola, 2002). While the kernel trick allows pairwise MMD computation without explicitly constructing the features (Gretton et al., 2012), it can only produce $\mathcal{O}(n^2)$ distances and does not yield explicit embeddings. To overcome this, we approximate the Gaussian kernel using random Fourier features (Rahimi and Recht, 2007), which provide a $D$-dimensional mapping that approximates the infinite dimensional feature space.

Concretely, given state $\mathbf{s} \in \mathcal{S}$ and action $\mathbf{a} \in \mathcal{A}$, we define a $D$-dimensional random feature map

$$\phi(\mathbf{s}, \mathbf{a}) = \sqrt{\frac{2}{D}} \left[\cos(\mathbf{w}_1^T[\mathbf{s}; \mathbf{a}] + \mathbf{b}_1), \ldots, \cos(\mathbf{w}_D^T[\mathbf{s}; \mathbf{a}] + \mathbf{b}_D)\right], \quad (3)$$

where $\mathbf{w}_i \sim \mathcal{N}(0, \sigma^{-2}I)$, $\mathbf{b}_i \sim \mathcal{U}(0, 2\pi)$, and $[\mathbf{s}; \mathbf{a}]$ denotes the concatenation of state and action vectors. The kernel width $\sigma$ determines the scale at which state-action pairs are considered similar.

Consider a policy $\pi$ with occupancy measure $\rho^\pi$. With a slight abuse of notation, let $\phi^\pi$ denote the embedding of $\pi$, defined as the empirical mean of the random Fourier features of $n$ i.i.d. samples from $\rho^\pi$. That is, $\phi^\pi = \frac{1}{n} \sum_j \phi(\mathbf{s}_j^\pi, \mathbf{a}_j^\pi)$ where $(\mathbf{s}_1^\pi, \mathbf{a}_1^\pi), \ldots, (\mathbf{s}_n^\pi, \mathbf{a}_n^\pi)$ are i.i.d. samples from $\rho^\pi$. This embedding (which we refer to as *policy embedding*) approximates the expected feature map under the policy's occupancy measure. The $\ell_2$ distance between embeddings of two policies approximates their behavioral difference as measured by MMD of their occupancy measures:

$$\|\phi^{\pi_1} - \phi^{\pi_2}\| \approx \mathrm{MMD}(\rho^{\pi_1}, \rho^{\pi_2}) \tag{4}$$

The quality of this approximation is characterized by the following theorem:

**Theorem 1** (MMD Approximation). *For any two policies $\pi_1, \pi_2$ with occupancy measure $\rho_1, \rho_2$ and embeddings $\phi_1, \phi_2$ estimated by taking the mean of the $D$ dimensional random Fourier features of $n$ i.i.d. samples from each occupancy measure,*

$$\Pr\left[\left|\|\phi_1 - \phi_2\|_2 - \mathrm{MMD}(\rho_1, \rho_2)\right| \geqslant \frac{3}{4}\varepsilon\right] \leqslant 2e^{-nc\varepsilon^2} + \mathcal{O}\left(\frac{1}{\varepsilon^2} \exp\left(\frac{-D\varepsilon^2}{64(d+2)}\right)\right) + 6e^{-\frac{n\varepsilon^2}{8}}, \tag{5}$$

*where $d$ is the dimension of state-action vectors and $c > 0$ is a constant. A proof is provided in Appendix A.*

This theorem establishes that the distance between our embeddings, $\|\phi_1 - \phi_2\|$, reliably approximates the true MMD between occupancy measures with high probability. Hence, the geometry of occupancy measures is captured by their embeddings. The approximation error is controlled by the number of samples $n$ and the embedding dimension $D$. Importantly, the state-action dimension $d$ appears only once in the denominator of an exponential term, suggesting that scaling to more complex domains requires $D$ to grow only linearly with $d$.

We should also mention a subtlety regarding the practical computation of the policy embedding $\phi^\pi$. To compute $\phi^\pi$, we need i.i.d samples from the occupancy measure $\rho^\pi$. We can obtain these by collecting $n$ independent rollouts of $\pi$ and selecting one state-action pair from each trajectory according to a Geometric distribution with parameter $1 - \gamma$. However, this leads to very inefficient use of the collected data as it discards all but one transition from each trajectory. Therefore, in practice, we use $\psi^\pi$ as defined in Eq. 6 instead of $\phi^\pi$ as the policy embedding. Intuitively, this is justified by noting that $\psi^\pi$ has the same expectation as as $\phi^\pi$ but leverages all collected transitions, which potentially reduces the variance. More concretely, we show in Appendix B that the distance between these $\psi$ embeddings also approximates the true MMD, with an approximation error that decays exponentially in $n$ and $D$, albeit at a different rate. Consequently, we use $\psi^\pi$ to denote the policy embedding from this point on.

$$\psi^\pi = \frac{1}{n} \sum_{j=1}^{n} (1 - \gamma) \sum_{t=0}^{T} \gamma^t \phi(\mathbf{s}_t^j, \mathbf{a}_t^j) \tag{6}$$

## 3.2 The AutoQD algorithm

Given policy embeddings that encode behavioral differences, we project them into a low-dimensional space (with $k \ll D$ dimensions) to serve as behavior descriptors for QD optimization. As explained in Sec. 2.2, this is needed because QD algorithms discretize each dimension of the behavior space, yielding an archive that grows exponentially with dimension. We perform this projection using an affine transformation $\mathrm{desc}(\pi) = \mathbf{A}\psi^\pi + \mathbf{b}$. The parameters of this transformation, $\mathbf{A} \in \mathbb{R}^{k \times D}, \mathbf{b} \in \mathbb{R}^k$, are derived by performing *Calibrated Weighted PCA (cwPCA)*, on the embeddings of policies in the archive. cwPCA makes some small modifications to PCA (F.R.S., 1901) to make it more suitable for the specification of behavior descriptors. In particular, it applies PCA to policy embeddings *after weighting them by their fitness*, so that better policies have greater influence on the principal directions. This biases the components toward capturing behavior variation among better policies, encouraging exploration among high-quality behaviors. Following this, we apply a simple *calibration* step: we scale each output axis so that most projected embeddings lie in the range $[-1, 1]$. This ensures stable and fixed archive bounds throughout the algorithm. Appendix C provides full details, including the precise form of the affine map, additional motivation, and an ablation study on the effect of the weighting mechanism.

Putting these pieces together, Algorithm 1 presents our method in its entirety. AutoQD combines the BDs described above with CMA-MAE to discover diverse and high-performing policies. It alternates between: (1) using the current descriptors to discover diverse policies with QD optimization, and (2) refining the descriptors based on the expanded archive of policies. For clarity and conciseness, Algorithm 1 abstracts the internal mechanics of CMA-MAE, omitting details of its initialization and update step. Detailed pseudocodes for these components are provided in Appendix D.

---

**Algorithm 1** AutoQD

---

1: **Input:** MDP $(\mathcal{S}, \mathcal{A}, P, R, \gamma)$, embedding dimension $D$, behavioral descriptor dimension $k$, number of iterations $n$, Update schedule $\{t_1, t_2, \ldots\}$
2: **Output:** Archive of diverse and high-performing policies $\mathbb{A}$
3: **Initialize:**
      CMA-MAE archive and parameters: $\mathbb{A}, \text{QDState} \leftarrow \texttt{CMA\_MAE\_Init}(k)$
      Affine map parameters: $\mathbf{A}, \mathbf{b}$
4: Sample random features $\{\mathbf{w}_i\}_{i=1}^{D} \sim \mathcal{N}(0, \sigma^{-2}I)$ and offsets $\{\mathbf{b}_i\}_{i=1}^{D} \sim \mathcal{U}(0, 2\pi)$
5: **for** $t \in \{1, 2, \ldots, n\}$ **do**
6:     **if** $t \in \{t_1, t_2, \ldots\}$ **then**                   ▷ Time to update descriptors
7:         $\Psi = [\psi^{\pi_1}, \ldots, \psi^{\pi_m}]$ for $\pi_i \in \mathbb{A}$       ▷ Policy embeddings as defined in Eq. 6
8:         $\mathbf{A}, \mathbf{b} \leftarrow \text{cwPCA}(\Psi, k)$
9:         Update behavioral descriptors: $\text{desc}(\pi) = \mathbf{A}\psi^{\pi} + \mathbf{b}$
10:     **end if**
11:     $\mathbb{A}, \text{QDState} \leftarrow \texttt{CMA\_MAE\_Step}(\mathbb{A}, \text{QDState}, \text{desc})$    ▷ Perform one step of QD optimization
12: **end for**
13: **return** final archive $\mathbb{A}$

---

## 4 EXPERIMENTS

To empirically validate the effectiveness of AutoQD in discovering diverse and high-performing behaviors, we evaluated it on six standard continuous control tasks from the Gymnasium library (Towers et al., 2024), including five from the widely-used MuJoCo benchmark suite (Todorov et al., 2012). These environments are standard benchmarks for RL and remain challenging for many evolutionary approaches, despite recent progress in the field.

### 4.1 BASELINES

We compare our method to five baselines that have demonstrated strong performance in prior work and represent distinct strategies for obtaining diverse and high-quality populations.

**RegularQD** applies a standard QD algorithm using hand-crafted BDs specific to each environment.
**Aurora** (Grillotti and Cully, 2022a) learns a behavior space by training an autoencoder on the visited states and uses the latent encoding of the last state in a rollout of the policy as the BD.
**LSTM-Aurora** (Chalumeau et al., 2023) extends AURORA by using LSTMs to encode full trajectories and using the hidden state of the encoder LSTM as the behavioral descriptor.
**DvD-ES** (Parker-Holder et al., 2020) employs evolutionary strategies to jointly optimize a population of policies for both task performance and diversity.
**SMERL** (Kumar et al., 2020) is an RL-based algorithm that trains a skill-conditioned policy using Soft Actor-Critic (Haarnoja et al., 2018) and uses an additional reward derived from a discriminator to encourage diversity among skills.

We use CMA-MAE (Fontaine and Nikolaidis, 2023) for all QD methods due to its simplicity and robustness across tasks. Additionally, following Choromanski et al. (2018), we use Toeplitz matrices to parameterize the policies for these methods to reduce parameters and improve the performance of CMA-MAE. The full set of hyperparameters and more discussion about the implementation details are provided in Appendix E.

### 4.2 EVALUATION METRICS

To ensure a fair comparison, we employ three main metrics: the Ground-Truth QD Score (GT QD Score), the Vendi Score (VS), and the Quality-Weighted Vendi Score (qVS).

Table 1: Comparison of AutoQD and baseline methods across six environments. Each environment is evaluated using GT QD Score (QD) reported in units of $10^4$ for readability, qVS, and VS metrics. Reported values are the mean $\pm$ standard error over evaluations with three different random seeds. Higher values indicate better performance for all metrics.

| Metric | AutoQD | RegularQD | Aurora | LSTM-Aurora | DvD-ES | SMERL |
|---|---|---|---|---|---|---|
| **Ant** | | | | | | |
| QD ($\times 10^4$) | **361.43 $\pm$ 2.17** | 182.58 $\pm$ 2.53 | 5.57 $\pm$ 1.48 | 19.24 $\pm$ 1.1 | 0.29 $\pm$ 0.1 | 1.02 $\pm$ 0.23 |
| qVS | **60.23 $\pm$ 9.4** | 39.35 $\pm$ 3.99 | 0.56 $\pm$ 0.01 | 1.11 $\pm$ 0.41 | 0.49 $\pm$ 0.00 | 0.97 $\pm$ 0.15 |
| VS | **72.37 $\pm$ 10.63** | 39.49 $\pm$ 3.93 | 1.11 $\pm$ 0.01 | 1.9 $\pm$ 0.54 | 1.00 $\pm$ 0.00 | 1.29 $\pm$ 0.18 |
| **HalfCheetah** | | | | | | |
| QD ($\times 10^4$) | **30.78 $\pm$ 2.72** | 24.91 $\pm$ 3.43 | 11.35 $\pm$ 4.69 | 11.38 $\pm$ 2.02 | 0.85 $\pm$ 0.23 | 1.61 $\pm$ 0.37 |
| qVS | 1.35 $\pm$ 0.6 | 2.07 $\pm$ 0.13 | **2.39 $\pm$ 0.42** | 1.71 $\pm$ 0.21 | 1.15 $\pm$ 0.09 | 1.78 $\pm$ 0.51 |
| VS | 5.29 $\pm$ 1.59 | 3.44 $\pm$ 0.34 | **5.8 $\pm$ 0.81** | 4.83 $\pm$ 0.16 | 1.19 $\pm$ 0.11 | 3.55 $\pm$ 0.56 |
| **Hopper** | | | | | | |
| QD ($\times 10^4$) | **1.84 $\pm$ 0.29** | 1.2 $\pm$ 0.03 | 1.06 $\pm$ 0.09 | 1.36 $\pm$ 0.01 | 0.56 $\pm$ 0.18 | 0.97 $\pm$ 0.15 |
| qVS | **1.94 $\pm$ 0.04** | 1.35 $\pm$ 0.05 | 0.66 $\pm$ 0.09 | 0.36 $\pm$ 0.08 | 0.9 $\pm$ 0.32 | 1.81 $\pm$ 0.22 |
| VS | **4.5 $\pm$ 0.2** | 2.85 $\pm$ 0.04 | 2.67 $\pm$ 0.09 | 2.13 $\pm$ 0.29 | 1.27 $\pm$ 0.13 | 3.34 $\pm$ 0.24 |
| **Swimmer** | | | | | | |
| QD ($\times 10^4$) | **21.31 $\pm$ 4.57** | 11.09 $\pm$ 0.08 | 8.05 $\pm$ 0.58 | 10.26 $\pm$ 0.72 | 0.22 $\pm$ 0.02 | 0.02 $\pm$ 0.00 |
| qVS | **6.04 $\pm$ 0.66** | 3.17 $\pm$ 0.19 | 3.09 $\pm$ 0.15 | 3.82 $\pm$ 0.77 | 1.16 $\pm$ 0.1 | 0.24 $\pm$ 0.06 |
| VS | **16.92 $\pm$ 3.68** | 4.67 $\pm$ 0.35 | 6.75 $\pm$ 0.25 | 7.21 $\pm$ 1.95 | 1.2 $\pm$ 0.13 | 2.16 $\pm$ 0.57 |
| **Walker2d** | | | | | | |
| QD ($\times 10^4$) | **18.36 $\pm$ 2.58** | 11.39 $\pm$ 0.55 | 7.71 $\pm$ 1.26 | 12.99 $\pm$ 0.77 | 0.61 $\pm$ 0.11 | 1.17 $\pm$ 0.14 |
| qVS | 7.22 $\pm$ 2.08 | **9.08 $\pm$ 0.53** | 1.11 $\pm$ 0.08 | 2.12 $\pm$ 0.07 | 1.47 $\pm$ 0.26 | 2.74 $\pm$ 0.42 |
| VS | 8.4 $\pm$ 3.2 | **10.17 $\pm$ 0.89** | 2.5 $\pm$ 0.13 | 4.17 $\pm$ 0.47 | 1.58 $\pm$ 0.29 | 3.2 $\pm$ 0.17 |
| **BipedalWalker** | | | | | | |
| QD ($\times 10^4$) | **6.09 $\pm$ 0.22** | 1.81 $\pm$ 0.02 | 3.0 $\pm$ 0.2 | 3.36 $\pm$ 0.08 | 0.09 $\pm$ 0.03 | 0.14 $\pm$ 0.01 |
| qVS | **5.16 $\pm$ 0.17** | 0.81 $\pm$ 0.02 | 1.12 $\pm$ 0.08 | 1.67 $\pm$ 0.34 | 1.03 $\pm$ 0.02 | 2.11 $\pm$ 0.27 |
| VS | **12.17 $\pm$ 0.52** | 1.57 $\pm$ 0.03 | 2.88 $\pm$ 0.21 | 3.36 $\pm$ 0.46 | 1.06 $\pm$ 0.00 | 5.54 $\pm$ 0.42 |

**GT QD Score** is the QD score of a population when its solutions are inserted into an archive that uses hand-designed BDs. It evaluates the quality and diversity of a population using expert-defined behavior spaces. These are the same BDs that the RegularQD baseline uses and are commonly employed in prior work. **Vendi Score (VS)** (Friedman and Dieng, 2023) quantifies a population's diversity based on pairwise similarities between their occupancy embeddings. Given a population of size $n$ and a positive-definite kernel matrix $K \in \mathbb{R}^{n \times n}$ where $K_{ij} \in [0, 1]$ is the similarity of the $i$-th and $j$-th members of the population, the Vendi Score is defined as $\mathrm{VS}(K) = \exp\left(-\sum_{i=1}^{n} \bar{\lambda}_i \log \bar{\lambda}_i\right)$, where $\bar{\lambda}_1, \bar{\lambda}_2, \ldots, \bar{\lambda}_n$ are the normalized eigenvalues of $K$ (i.e., they sum to one). Importantly, VS measures the *effective population size* and enables comparison between populations of varying sizes, as is the case with our baselines. Lastly, **Quality-Weighted Vendi Score (qVS)** (Nguyen and Dieng, 2024) extends the VS by incorporating solution quality: $\mathrm{qVS}(K) = \left(\frac{1}{n} \sum_{i=1}^{n} J(\pi_i)\right) \mathrm{VS}(K)$, where $J(\pi_i)$ is the fitness of the $i$-th individual, $\pi_i$. Since, qVS requires all objectives to be positive, we scale all objectives to the $[0, 1]$ range by adding a constant offset and dividing each return by the highest mean return achieved by any of the algorithms in that environment, prior to computing qVS.

To construct the kernel matrix $K$ used by VS and qVS, we use a Gaussian kernel applied to the inner product of the Random Fourier Feature (RFF) embeddings of policies. Although these embeddings are structurally similar to those used by AutoQD, we employ a separate, larger set of RFFs solely for evaluation to ensure a fair comparison. Our choice of embeddings is motivated by our theoretical results showing that distances between these embeddings asymptotically reflect distances between policy occupancy measures. For a more detailed analysis of qVS and its theoretical properties, we refer the reader to Nguyen and Dieng (2024).

## 4.3 MAIN RESULTS: POLICY DISCOVERY

Table 1 compares AutoQD with the baseline algorithms across six tasks. For each combination, we report the mean and standard error across three random seeds. AutoQD consistently outperforms

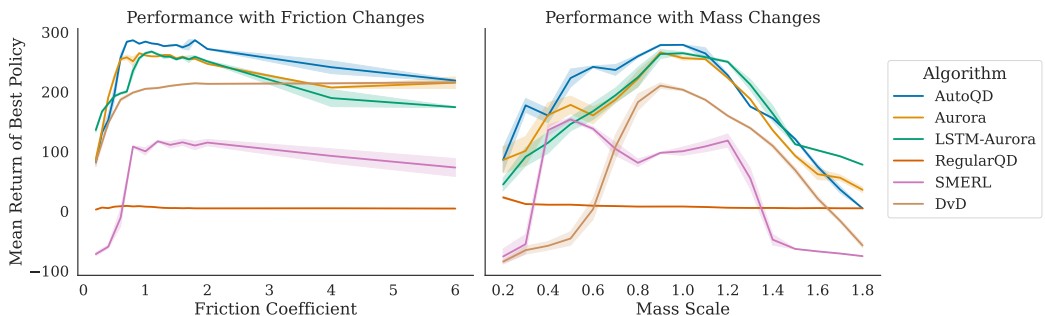

Figure 3: Performance of the best policy found by each algorithm under changing friction (left) or mass scale (right). The shaded regions represent the standard error across 32 evaluation seeds.

the baselines in most environments; The only exceptions being the **Walker2d** and **HalfCheetah** environments, where the best qVS and VS are achieved by RegularQD and Aurora, respectively.

In HalfCheetah, AutoQD was able to discover diverse policies, but the policies tended to be relatively low performing, reflected by its high VS and low qVS. Visual inspection showed that AutoQD discovered many policies that moved forward by "sliding" via subtle joint movements. While these behaviors were novel and diverse, they resulted in slow movement, and as a result, lower overall rewards. In Walker2d, AutoQD seemingly overemphasized the role of the bottom-most (feet) joints, missing out on interesting behavioral variations that could be achieved, for instance, by fully lifting the legs. Nevertheless, AutoQD ranked second in this domain, outperforming all other baselines. Appendix F provides more fine-grained statistics and further analysis of AutoQD's lower performance in these two domains. Moreover, Appendix J presents qualitative analysis and visualizations of the behaviors discovered by AutoQD.

## 4.4 APPLICATION: ADAPTATION TO DIFFERENT DYNAMICS

A key motivation for discovering diverse populations is adaptability, since a collection of behaviorally diverse policies is more likely to include one that performs well under altered environment conditions. To test this, we evaluated populations from AutoQD and the baselines in the Bipedal-Walker environment with two types of variations: scaling the friction coefficient and altering the robot mass. Figure 3 shows the performance of each method's best policy under these changes, and their area under the curve (AUC) provides a scalar measure of robustness, with higher AUC indicating greater adaptability. As Table 2 shows, AutoQD's population maintains relatively high performance across both variations, and achieves the highest AUC.

Beyond measuring the performance of the single best policy, it is also helpful to analyze the performance of the top-performing subset and even the full distribution of returns across all policies within a population. To quantify this, we counted the number of policies in each population that maintained a significant fraction $0 < p < 1$ of the performance achieved by the best overall policy found in the original, unaltered environment. Specifically, if $R$ represents the highest return achieved by any policy across all populations in the original environment, we counted policies whose mean return was at least $Rp$. Intuitively, this would reflect the number of policies that successfully adapt to the environmental changes by maintaining a high reward.

Figure 4 illustrates the count of these "successful" policies across different friction coefficients for two success thresholds: $p = 0.9$ and $p = 0.7$. At the strict threshold of $p = 0.9$, AutoQD's population consistently includes a larger number of successfully adapted policies compared to the baselines. As we loosen the success criteria to $p = 0.7$, the number of successful policies increases across the board. Notably, the population of LSTM-Aurora is shown to contain many successful policies when the friction coefficient is in the range $[1, 3]$. However, as the friction coefficient increases further, both AutoQD and DvD-ES prove to be more capable of finding successful policies. We observe similar general trends when varying the robot mass scale, with additional plots and the full distribution of returns provided in Appendix G. Overall, these findings show that AutoQD's generated population not only contains a single policy that can adapt to changing environmental

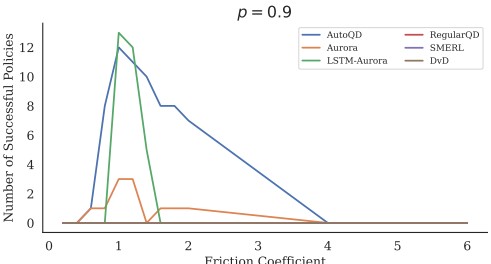 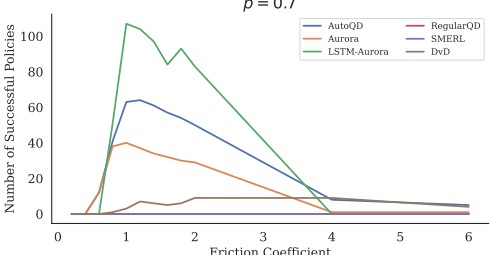

Figure 4: Number of successfully adapting policies in each population under changing friction. A policy is considered successful if its mean return is at least $Rp$, where $R$ is the highest overall return achieved in the unaltered environment. Results are shown for two success thresholds: $p = 0.9$ (left) and $p = 0.7$ (right).

| Metric | AutoQD | RegularQD | Aurora | LSTM-Aurora | DvD-ES | SMERL |
|---|---|---|---|---|---|---|
| Friction AUC | **1429.66** | 30.27 | 1309.41 | 1226.29 | 1204.03 | 496.23 |
| Mass AUC | **295.65** | 12.8 | 260.60 | 271.83 | 113.68 | 71.38 |

Table 2: Comparison of Area Under the Curve (AUC) for each algorithm across friction and mass variations. Higher values indicate better adaptability to changing parameters.

conditions, but also includes many (either the most or the second most among the baselines) policies that demonstrate substantial adaptability.

## 5 RELATED WORK

**Quality-Diversity methods.** Quality-Diversity algorithms discover collections of solutions that balance performance and diversity across specified behavioral dimensions (Pugh et al., 2016). MAP-Elites (Cully et al., 2015b) pioneered this approach by maintaining an archive of solutions organized by their behavioral characteristics. CMA-ME (Fontaine et al., 2020) reformulated the QD problem as single objective optimization, enabling the use of powerful blackbox optimization methods like CMA-ES (Hansen, 2016) instead of relying solely on random mutations. The more recent CMA-MAE (Fontaine and Nikolaidis, 2023), which is used as the backbone QD algorithm in this paper, further improved this method by introducing the idea of *soft archives*. More recently, gradient-based variants like DQD (Fontaine and Nikolaidis, 2021), PGA-MAP-Elites (Nilsson and Cully, 2021) and PPGA (Batra et al., 2024) have made further progress by leveraging policy gradients.

**Unsupervised QD approaches.** Most prior work such as Aurora (Grillotti and Cully, 2022a), LSTM-Aurora (Chalumeau et al., 2023), and TAXONS (Paolo et al., 2020) learn behavioral descriptors by training autoencoders on states, relying on the hypothesis that representations capturing state information also reflect policy behavior. Grillotti and Cully (2022b) argues in favor of this approach by showing that the entropy of the encoded trajectories lower-bounds the entropy of the full trajectories. However, their analysis assumes a discrete state space and does not formally link trajectory entropy to policy diversity. In contrast, AutoQD's embeddings are based on policy-induced occupancy measures, offering a theoretically grounded representation of behavior.

**Unsupervised RL for skill discovery.** RL community has explored related approaches for learning diverse behaviors. DIAYN (Eysenbach et al., 2019) maximizes mutual information between skills and states, encouraging skills to visit distinct regions of the state space without using reward signals. DADS (Sharma et al., 2020) extends this by maximizing mutual information between skills and transitions, favoring predictable outcomes. However, both methods ignore the task reward. SMERL (Kumar et al., 2020) and DoMiNo (Zahavy et al., 2023) incorporate task rewards into diversity objectives. SMERL directly augments DIAYN's objective with task rewards, while DoMiNo frames the problem as a constrained MDP, maximizing diversity by encouraging distance between

state occupancies of near-optimal policies. Both highlight the benefits of diverse, high-performing policies but require a fixed number of skills and tend to scale poorly with skill count. In this work, we compared our method with SMERL, as its open-source implementation is readily available.

**Policy embedding and representation.** In a middle ground between QD methods and unsupervised RL approaches, DvD (Parker-Holder et al., 2020) characterizes policies through their actions in (random) set of states, resembling the off-policy embeddings from Pacchiano et al. (2020). However, these embeddings lack the theoretical backing that our method provides. Furthermore, like SMERL, their proposed algorithm requires specifying the number of policies in advance and faces stability issues as this number increases. Chen et al. (2023) also share conceptual similarities with our approach, though in the context of transfer learning. They learn a Q-function basis by training policies on features from randomly initialized networks. In contrast, we use random Fourier features to embed occupancy measures directly, enabling QD optimization without prior RL training.

Our use of Random Fourier Features (Rahimi and Recht, 2007) to embed occupancy measures connects to theoretical work on kernel approximations (Rudi and Rosasco, 2017; Rahimi and Recht, 2008). A key insight of our approach is recognizing that these techniques can be applied to represent policy behaviors in a theoretically principled way. By embedding occupancy measures and applying dimensionality reduction, we automatically generate behavioral descriptors that capture essential policy characteristics without manual specification.

## 6 CONCLUSION

We introduced **AutoQD**, a novel approach for applying Quality-Diversity (QD) optimization to sequential decision-making tasks without handcrafted behavior descriptors. By embedding policies based on their occupancy measures and projecting to a compact behavior space, AutoQD can be integrated with CMA-MAE and achieves strong empirical performance.

**Limitations.** AutoQD has several limitations. First, in highly stochastic environments, accurately estimating policy embeddings may require many trajectories, which reduces sample efficiency. Second, as discussed in Appendix F, when the behavior descriptor is low-dimensional, exploration may concentrate on stable yet simple behaviors, hindering the discovery of more complex ones. In addition, the choice of kernel bandwidth influences the sensitivity of the embeddings; while we use a fixed bandwidth in this work, dynamically adapting it during training could allow the embeddings to better capture behavioral distinctions at different stages of learning. In this study, we used AutoQD with CMA-MAE because of its simplicity and stability. However, AutoQD is in principle compatible with any standard QD algorithm. As a result, it inherits the scalability challenges of existing QD optimizers, particularly with large policy networks and high-dimensional behavior spaces (Tjanaka et al., 2023a), but can also benefit directly from future advances in QD algorithm design. Finally, although QD methods promote behavioral diversity, they may fall short of RL methods in pure reward optimization. Nevertheless, we expect this gap to narrow as QD algorithms continue to improve.

**Future Work.** A promising direction involves integrating AutoQD with gradient-based QD methods such as PGA-ME (Nilsson and Cully, 2021) and PPGA (Batra et al., 2024). While these methods typically offer better performance, their training objectives can become unstable as a result of the iterative refinement of the behavior space by AutoQD. By identifying the sources of these instabilities and mitigating them, future work can improve performance and increase sample efficiency. Furthermore, extending AutoQD to environments with image-based observations is also a direction worth pursuing and could unlock exciting capabilities for autonomous agents. Lastly, the policy embeddings produced by AutoQD could find applications beyond QD, including open-ended learning, imitation learning, and inverse RL. They may also prove useful for analyzing learned policies, for example through clustering and other forms of characterization.

## 7 ACKNOWLEDGMENTS AND DISCLOSURE OF FUNDING

We would like to thank Varun Bhatt, Sophie Hsu, Aaquib Tabrez, Bryon Tjanaka, and Shihan Zhao for their feedback on a preliminary version of this work, as well as Saba Hashemi for assistance

with the design of the visualizations. This work has been partially supported by the NSF CAREER #2145077, NSF NRI #2024949 and the DARPA EMHAT project.

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

## A PROOF OF THEOREM 1

As our main result, we show that the $\ell_2$ distance between our embeddings of occupancy measures (estimated from samples) is a good approximation of the true MMD between occupancy measures. Formally, let $P$ and $Q$ be two occupancy measures defined over $d$ dimensional state-action vectors. Let $k$ be the Gaussian kernel and $\phi : \mathbb{R}^d \to \mathbb{R}^D$ be the random Fourier features that map state-action vectors to a $D$ dimensional embedding space. Given $n$ samples $\{x_1, \cdots x_n\}$ from $P$ we define $\phi_P = \frac{1}{n} \sum_i \phi(x_i)$ as the embedding of $P$. Similarly, we define $\phi_Q$ as the embedding of $Q$ obtained from $n$ samples $y_1, \cdots, y_n$. The claim is that $\|\phi_P - \phi_Q\|_2$ is a good approximation of $\mathrm{MMD}(P, Q)$. The following are the steps we take to complete the proof.

1. We start by showing that with high probability $\mathrm{MMD}^2(P, Q)$ and $\|\phi_P - \phi_Q\|_2^2$ are close to one another. This is done in four steps where we

   (a) show that $\|\phi_P - \phi_Q\|_2^2$ is close to $\widetilde{\mathrm{MMD}}$,

   (b) show that $\widetilde{\mathrm{MMD}}$ is close to $\widehat{\mathrm{MMD}}$,

   (c) show that $\widehat{\mathrm{MMD}}$ is close to $\mathrm{MMD}^2$,

   (d) conjoin the previous three bounds to show that $\|\phi_P - \phi_Q\|_2^2$ is close to $\mathrm{MMD}^2$.

2. Then, we show that $\mathrm{MMD}(P, Q)$ is close to $\|\phi_P - \phi_Q\|_2$.

Recall that from the definition of MMD and using the kernel trick we have

$$\mathrm{MMD}^2(P, Q) = \mathbb{E}_{X, X' \sim P}[k(X, X')] + \mathbb{E}_{Y, Y' \sim Q}[k(Y, Y')] - 2\mathbb{E}_{X \sim P, Y \sim Q}[k(X, Y)]. \quad (7)$$

Let us start from $\|\phi_P - \phi_Q\|_2^2$ and step-by-step get closer to the quantity above. We have

$$\|\phi_P - \phi_Q\|_2^2 = (\phi_P - \phi_Q)^T (\phi_P - \phi_Q) \quad (8)$$

$$= \phi_P^T \phi_P + \phi_Q^T \phi_Q - 2\phi_P^T \phi_Q. \quad (9)$$

Examining each of these three terms more carefully, we see that

$$\phi_P^T \phi_P = \frac{1}{n^2} \sum_{i,j} \phi(x_i)^T \phi(x_j) = \frac{1}{n^2} \sum_i \phi(x_i)^T \phi(x_i) + \frac{1}{n^2} \sum_{i \neq j} \phi(x_i)^T \phi(x_j), \quad (10)$$

$$\phi_Q^T \phi_Q = \frac{1}{n^2} \sum_{i,j} \phi(y_i)^T \phi(y_j) = \frac{1}{n^2} \sum_i \phi(y_i)^T \phi(y_i) + \frac{1}{n^2} \sum_{i \neq j} \phi(y_i)^T \phi(y_j), \quad (11)$$

$$\phi_P^T \phi_Q = \frac{1}{n^2} \sum_{i,j} \phi(x_i)^T \phi(y_j). \quad (12)$$

Now, let $\widetilde{\mathrm{MMD}}$ be defined as

$$\widetilde{\mathrm{MMD}} = \frac{1}{n} \sum_{i=1}^n \phi(x_i)^T \phi(x_i') + \frac{1}{n} \sum_{i=1}^n \phi(y_i)^T \phi(y_i') - \frac{2}{n} \sum_{i=1}^n \phi(x_i)^T \phi(y_i), \quad (13)$$

where $x_i, x_i'$'s are i.i.d. samples from $P$ and $y_i, y_i'$ are i.i.d. samples from $Q$. From this, we can see that the expectations of $\|\phi_P - \phi_Q\|_2^2$ and $\widetilde{\mathrm{MMD}}$ are quite similar and in fact, they are the same at the limit of $n \to \infty$.

$$\mathbb{E}\left[\|\phi_P - \phi_Q\|_2^2\right] = \left(\frac{n-1}{n}\right) \mathbb{E}\left[\widetilde{\mathrm{MMD}}\right] + \frac{1}{n} \mathbb{E}_{X \sim P, Y \sim Q}\left[\|\phi(X)\|^2 + \|\phi(Y)\|^2\right]. \quad (14)$$

Using the fact that, by the definition of random Fourier features, the entries of $\phi(x)$ are bounded in $[-\frac{\sqrt{2}}{\sqrt{D}}, \frac{\sqrt{2}}{\sqrt{D}}]$ we see that the difference between $\mathbb{E}\left[\|\phi_P - \phi_Q\|_2^2\right]$ and $\mathbb{E}\left[\widetilde{\mathrm{MMD}}\right]$ is at most $\mathcal{O}(\frac{1}{n})$. Similarly, for any $x, x'$, we see that $\phi(x)^T \phi(x') \in [-2, 2]$. Therefore, each of the $n$ summands of $\widetilde{\mathrm{MMD}}$ take values in $[-8, 8]$.

Now, an application of Hoeffding's inequality yields that with probability at least $1 - \delta$:

$$\left|\widehat{\mathrm{MMD}} - \mathbb{E}\left[\widehat{\mathrm{MMD}}\right]\right| \leqslant 16\sqrt{\frac{\log\frac{2}{\delta}}{2n}}. \tag{15}$$

Similarly, $\left|\|\phi_P - \phi_Q\|^2\right.$ is a function of $2n$ independent samples and satisfies a bounded difference property: changing a single sample changes its value by at most $c = \frac{16}{n} + \frac{8}{n^2}$ which is at most $\frac{24}{n}$ for $n \geqslant 1$. Therefore, an application of McDiarmid's inequality (McDiarmid et al., 1989) shows that with probability at least $1 - \delta$,

$$\left|\|\phi_P - \phi_Q\|^2 - \mathbb{E}\left[\|\phi_P - \phi_Q\|^2\right]\right| \leqslant 24\sqrt{\frac{\log\frac{2}{\delta}}{n}}, \tag{16}$$

Combining these with the triangle inequality and using the union bound we get that with probability at least $1 - 2\delta$

$$\left|\|\phi_P - \phi_Q\|^2 - \widehat{\mathrm{MMD}}\right| \leqslant 48\sqrt{\frac{\log\frac{2}{\delta}}{n}} + \mathcal{O}(\frac{1}{n}). \tag{17}$$

For large values of $n$ the first term on the right hand side dominates, therefore we can say that with probability at least $1 - 2\delta$

$$\left|\|\phi_P - \phi_Q\|^2 - \widehat{\mathrm{MMD}}\right| = \mathcal{O}\left(\sqrt{\frac{-\log\delta}{n}}\right). \tag{18}$$

Therefore, for some non-negative constant $c$ we have

$$\Pr\left[\left|\|\phi_P - \phi_Q\|^2 - \widehat{\mathrm{MMD}}\right| \geqslant \varepsilon\right] \leqslant 2e^{-nc\varepsilon^2} \tag{19}$$

We now move on to the next part of the proof. Define $\widetilde{\mathrm{MMD}}$ as follows.

$$\widetilde{\mathrm{MMD}} = \frac{1}{n}\sum_{i=1}^{n}k\left(x_i, x_i'\right) + \frac{1}{n}\sum_{i=1}^{n}k\left(y_i, y_i'\right) - \frac{2}{n}\sum_{i=1}^{n}k\left(x_i, y_i\right). \tag{20}$$

In words, $\widetilde{\mathrm{MMD}}$ is just like $\widehat{\mathrm{MMD}}$ but with all of the inner products of random Fourier features replaced by kernel operations. We can see that

$$|\widetilde{\mathrm{MMD}} - \widehat{\mathrm{MMD}}| = \left|\frac{1}{n}\sum_{i=1}^{n}\left[k\left(x_i, x_i'\right) - \phi(x_i)^T\phi(x_i')\right]\right. \tag{21}$$

$$+ \frac{1}{n}\sum_{i=1}^{n}\left[k\left(y_i, y_i'\right) - \phi(y_i)^T\phi(y_i')\right] \tag{22}$$

$$\left. - \frac{2}{n}\sum_{i=1}^{n}\left[k\left(x_i, y_i\right) - \phi(x_i)^T\phi(y_i)\right]\right|. \tag{23}$$

Next, we make use of the following lemma that guarantees the uniform convergence of Fourier features stated in section 3 of Rahimi and Recht (2007):

$$\Pr\left[\sup_{x,y}|\phi(x)^T\phi(y) - k(x,y)| \geqslant \varepsilon\right] \leqslant \mathcal{O}\left(\frac{1}{\varepsilon^2}\exp\left(-\frac{D\varepsilon^2}{4(d+2)}\right)\right) \tag{24}$$

where $d$ is the dimensionality of the state-action vectors. This implies that each of the terms (summands) in 23 is at most $\frac{\varepsilon}{4}$ with probability at least $1 - \mathcal{O}\left(\frac{16}{\varepsilon^2}\exp\left(-\frac{D\varepsilon^2}{64(d+2)}\right)\right)$. Substituting $\frac{\varepsilon}{4}$ in 23 and using the triangle inequality, we see that

$$|\widetilde{\mathrm{MMD}} - \widehat{\mathrm{MMD}}| \leqslant \frac{1}{n}\sum_{i=1}^{n}\frac{\varepsilon}{4} + \frac{1}{n}\sum_{i=1}^{n}\frac{\varepsilon}{4} + \frac{2}{n}\sum_{i=1}^{n}\frac{\varepsilon}{4} = \varepsilon, \tag{25}$$

with probability at least $1 - \mathcal{O}\left(\frac{16}{\varepsilon^2} \exp\left(-\frac{D\varepsilon^2}{64(d+2)}\right)\right)$. Therefore,

$$\Pr\left[\left|\widehat{\mathrm{MMD}} - \widetilde{\mathrm{MMD}}\right| \geqslant \varepsilon\right] \leqslant \mathcal{O}\left(\frac{16}{\varepsilon^2} \exp\left(\frac{-D\varepsilon^2}{64(d+2)}\right)\right). \tag{26}$$

This brings us to the third step of the proof where we connect $\widehat{\mathrm{MMD}}$ with $\mathrm{MMD}^2(P,Q)$. This is more straight forward to show, since each term in the former is the Monte Carlo estimate of the corresponding expectation in the latter. More formally,

$$\widehat{\mathrm{MMD}} - \mathrm{MMD}^2(P,Q) = \left(\frac{1}{n}\sum_{i=1}^{n} k\left(x_i, x_i'\right) - \mathbb{E}_P[k(X, X')]\right) \tag{27}$$

$$+ \left(\frac{1}{n}\sum_{i=1}^{n} k\left(y_i, y_i'\right) - \mathbb{E}_Q[k(Y, Y')]\right) \tag{28}$$

$$- 2\left(\frac{1}{n}\sum_{i=1}^{n} k\left(x_i, y_i\right) - \mathbb{E}_{P,Q}[k(X, Y)]\right). \tag{29}$$

Now note that in each of the three parentheses the first term is the empirical mean and the second term is the true mean. Combining this with the fact that $k(x, y)$ is always between 0 and 1, we can apply Hoeffding's inequality to each term to get a tail bound for each of them. For example, for the first parenthesis we get

$$\Pr\left(\left|\frac{1}{n}\sum_{i=1}^{n} k\left(x_i, x_i'\right) - \mathbb{E}_P[k(X, X')]\right| \geqslant \frac{\varepsilon}{4}\right) \leqslant 2\exp\frac{-n\varepsilon^2}{8}. \tag{30}$$

Applying the triangle inequality and the union bound we get

$$\Pr\left[\left|\widehat{\mathrm{MMD}} - \mathrm{MMD}^2(P,Q)\right| \geqslant \varepsilon\right] \leqslant 6\exp\frac{-n\varepsilon^2}{8} \tag{31}$$

Now, we can combine 19, 26, and 31 to bound the difference between $\|\phi_P - \phi_Q\|_2^2$ and $\mathrm{MMD}^2(P,Q)$. Note that by triangle inequality

$$\left|\|\phi_P - \phi_Q\|_2^2 - \mathrm{MMD}^2(P,Q)\right| \leqslant \left|\|\phi_P - \phi_Q\|_2^2 - \widetilde{\mathrm{MMD}}\right| + \left|\widetilde{\mathrm{MMD}} - \widehat{\mathrm{MMD}}\right| + \left|\widehat{\mathrm{MMD}} - \mathrm{MMD}^2\right|. \tag{32}$$

Combining the bounds that we have for each of the terms on the right hand side, we get

$$\Pr\left[\left|\|\phi_P - \phi_Q\|_2^2 - \mathrm{MMD}^2(P,Q)\right| \geqslant 3\varepsilon\right] \leqslant 2e^{-nc\varepsilon^2} + \mathcal{O}\left(\frac{1}{\varepsilon^2} \exp\left(\frac{-D\varepsilon^2}{64(d+2)}\right)\right) + 6e^{-\frac{n\varepsilon^2}{8}}. \tag{33}$$

This ensures that as we increase the number of samples $n$ and the number of features $D$, the probability of error decays exponentially.

Lastly, we shall derive a bound on $\left|\|\phi_P - \phi_Q\|_2 - \mathrm{MMD}(P,Q)\right|$. Note that

$$\left|\|\phi_P - \phi_Q\|_2^2 - \mathrm{MMD}^2(P,Q)\right| = \left|\|\phi_P - \phi_Q\|_2 - \mathrm{MMD}(P,Q)\right|\left(\|\phi_P - \phi_Q\|_2 + \mathrm{MMD}(P,Q)\right) \tag{34}$$

$$\leqslant \left|\|\phi_P - \phi_Q\|_2 - \mathrm{MMD}(P,Q)\right|(2+2) \tag{35}$$

$$\leqslant 4\left|\|\phi_P - \phi_Q\|_2 - \mathrm{MMD}(P,Q)\right|. \tag{36}$$

Replacing this back into the bound in 33 we get

$$\Pr\left[\left|\|\phi_P - \phi_Q\|_2 - \mathrm{MMD}(P,Q)\right| \geqslant \frac{3}{4}\varepsilon\right] \leqslant 2e^{-nc\varepsilon^2} + \mathcal{O}\left(\frac{1}{\varepsilon^2} \exp\left(\frac{-D\varepsilon^2}{64(d+2)}\right)\right) + 6e^{-\frac{n\varepsilon^2}{8}}. \tag{37}$$

Which is the result that we sought. This ensures that as we increase $n$ and $D$, the distances between the embeddings of occupancy measures reflect the true MMD distance between them with a high probability.

## B  JUSTIFICATION FOR THE EMPIRICAL TRAJECTORY EMBEDDINGS

As mentioned in Sec. 3.1, we use $\psi^\pi$ defined in Eq. 6 to compute the policy embeddings in practice. Here, we connect this choice to the MMD approximation Theorem 1 to show that using the practical, trajectory-based embeddings $\psi_P$ and $\psi_Q$ in place of the theoretically analyzed occupancy-measure embeddings $\phi_P$ and $\phi_Q$ (used in Appendix A) is theoretically justified. To do so, we show that, with high probability, the distance between $\psi_P$ and $\psi_Q$ is arbitrary close to that between $\phi_P$ and $\phi_Q$ for any two occupancy measures $P, Q$. Hence, all of the results in the main proof remain valid when we replace $\phi^\pi$ with $\psi^\pi$.

First, let us define the true mean feature vector as

$$\mu_P := \mathbb{E}_{(s,a)\sim P}[\phi(s,a)], \tag{38}$$

where $P$ is the occupancy measure of some policy $\pi$. The analysis in Appendix A uses i.i.d. samples $x_i \sim P$ and their empirical mean $\phi_P = \frac{1}{n}\sum_{i=1}^n \phi(x_i)$. However, in practice, we obtain feature vectors $z_i$ from trajectories $\tau_i$ by setting

$$z_i := (1-\gamma)\sum_{t=0}^{\infty}\gamma^t\phi(s_t^i, a_t^i), \tag{39}$$

and using their empirical mean $\psi_P = \frac{1}{n}\sum_{i=1}^n z_i$.

We will show that $|\|\psi_P - \psi_Q\|_2 - \|\phi_P - \phi_Q\|_2|$ is small with high probability. This guarantees that replacing $\phi$ by $\psi$ in the main proof does not affect any bounds, other than incurring an additional approximation error that will be derived below.

First, let us prove two simple lemmas regarding $\phi(x_i)$ and $z_i$ vectors.

**Lemma 1.** *Let $z = (1-\gamma)\sum_t \gamma^t\phi(s_t, a_t)$ be a feature vector obtained by sampling a trajectory from occupancy measure $P$ and let $x = \phi(s,a)$ where the state action pair $(s,a)$ is also sampled from $P$. Then, $\mathbb{E}_P[z] = \mathbb{E}_P[x]$.*

*Proof.* Note that $\mathbb{E}_P[x] = \mu_P$ by definition. Also,

$$\mathbb{E}[z] = (1-\gamma)\sum_{t=0}^{\infty}\gamma^t\mathbb{E}[\phi(S_t, A_t)] \tag{40}$$

$$= \sum_{s,a}\phi(s,a)\rho^\pi(s,a) \tag{41}$$

$$= \mu_P. \tag{42}$$

Therefore, $z$ and $x$ have the same expectation. $\square$

**Note (finite-horizon bias).** The derivation above assumed an infinite-horizon setting ($T = \infty$) so that a trajectory feature $z^\infty = (1-\gamma)\sum_{t=0}^{\infty}\gamma^t\phi(s_t, a_t)$ satisfies $\mathbb{E}[z^\infty] = \mu_P$. In practice we use truncated trajectories of length $T < \infty$. Defining $z^T = (1-\gamma)\sum_{t=0}^{T-1}\gamma^t\phi(s_t, a_t)$, the error (bias) between the infinite and finite-horizon features is

$$b_T = \mathbb{E}[z^\infty] - \mathbb{E}[z^T] = (1-\gamma)\sum_{t=T}^{\infty}\gamma^t\,\mathbb{E}[\phi(S_t, A_t)]. \tag{43}$$

Using $\|\phi(s,a)\|_2 \leqslant \sqrt{2}$ (following lemma) we obtain

$$\|b_T\|_2 \leqslant (1-\gamma)\sum_{t=T}^{\infty}\gamma^t\sqrt{2} = \sqrt{2}\gamma^T. \tag{44}$$

Thus, truncating trajectories at length $T$ introduces a deterministic bias of at most $\sqrt{2}\gamma^T$ in the trajectory features. Consequently, when comparing two occupancy measures $P, Q$ the truncation contributes at most $\|b_{T,P}\|_2 + \|b_{T,Q}\|_2 \leq 2\sqrt{2}\gamma^T$ to the total error and should be added to the sampling and RFF approximation terms in the final probabilistic bound (i.e., should be added to the $\frac{11}{4}\varepsilon$ term in the final bound 56). For simplicity, we consider the infinite horizon setting throughout the rest of the proof.

**Lemma 2.** *For any trajectory $\tau$ and its corresponding feature vector $z$ defined according to Equation 39, all coordinates of $z$ lie in $[-\frac{\sqrt{2}}{\sqrt{D}}, \frac{\sqrt{2}}{\sqrt{D}}]$.*

*Proof.* This follows immediately by noting that each coordinate of $\phi(s, a)$ is in the same range and each coordinate of $z$ is a geometric sum of these values. □

Hence, from the coordinate bound $|\phi_i(s, a)| \leqslant \sqrt{2/D}$ we have $\|\phi(s, a)\|_2 \leqslant \sqrt{2}$ and likewise $\|z\|_2 \leqslant \sqrt{2}$.

Now, we can define centered feature vectors $U_i := \phi(x_i) - \mu_P$ (for $\phi$-estimators) and $U_j := z_j - \mu_P$ (for $\psi$-estimators). For either case, using the triangle inequality yields that

$$\|U_i\|_2 \leqslant \|\phi(x_i)\|_2 + \|\mu_P\|_2 \leqslant 2\sqrt{2}, \tag{45}$$

$$\|U_j\|_2 \leqslant \|z_j\| + \|\mu_P\|_2 \leqslant 2\sqrt{2}. \tag{46}$$

We can now apply the following inequality which follows from Theorem 3.5 of Pinelis (2012):

**Lemma 3.** *Let $U_1, \ldots, U_n$ be independent, zero-mean random vectors in $\mathbb{R}^D$ satisfying $\|U_i\| \leqslant 2\sqrt{2}$ almost surely. Then, for any $r > 0$,*

$$\Pr\left[\|\frac{1}{n}\sum_{i=1}^{n} U_i\|_2 \geqslant r\right] \leqslant 2\exp\left(\frac{-nr^2}{16}\right). \tag{47}$$

*Proof.* Follows from applying Theorem 3.5 of Pinelis (2012) with increments $d_j = U_j$. Since $\|U_j\|_2 \leqslant 2\sqrt{2}$, we have $b_*^2 \leqslant \sum_{j=1}^{n}(2\sqrt{2})^2 = 8n$. Taking $D = 1$, the theorem yields the bound above. □

Setting $r = \frac{\varepsilon}{2}$ and using the centered feature vectors $U_i$ and $U_j$, this lemma shows that

$$\Pr\left[\|\phi_P - \mu_P\|_2 \geqslant \frac{\varepsilon}{2}\right] \leqslant 2\exp\left(\frac{-n\varepsilon^2}{64}\right), \tag{48}$$

$$\Pr\left[\|\psi_P - \mu_P\|_2 \geqslant \frac{\varepsilon}{2}\right] \leqslant 2\exp\left(\frac{-n\varepsilon^2}{64}\right). \tag{49}$$

By union bound, the probability that both deviations exceed $\varepsilon/2$ is at most $4\exp(-n\varepsilon^2/64)$. Therefore, with probability at least $1 - 4\exp(-n\varepsilon^2/64)$,

$$\|\phi_P - \psi_P\|_2 \leqslant \|\phi_P - \mu_P\|_2 + \|\psi_P - \mu_P\|_2 \leqslant \varepsilon. \tag{50}$$

Similarly, we can apply the same bound independently for another occupancy measure $Q$. Applying the union bound across both policies that with probability at least $1 - 8\exp(-n\varepsilon^2/64)$, both $\|\phi_P - \psi_P\|$ and $\|\phi_Q - \psi_Q\|$ are at most $\varepsilon$. But note that this would imply that

$$\left|\|\psi_P - \psi_Q\| - \|\phi_P - \phi_Q\|\right| \leqslant \|(\psi_P - \psi_Q) - (\phi_P - \phi_Q)\| \tag{51}$$

$$\leqslant \|\psi_P - \phi_P\| + \|\psi_Q - \phi_Q\| \tag{52}$$

$$\leqslant 2\varepsilon. \tag{53}$$

Putting everything together, we have shown

$$\Pr\left[|\|\psi_P - \psi_Q\| - \|\phi_P - \phi_Q\|| \geqslant 2\varepsilon\right] \leqslant 8e^{-nc\varepsilon^2} \tag{54}$$

with $c = \frac{1}{64}$. Combining this with the bound of Eq. 37 using the triangle inequality, we can see that $\|\psi_P - \psi_Q\|_2$ is close to $\mathrm{MMD}(P, Q)$, with high probability. More concretely, if we use a subscript of 1 for the constant $c$ in Eq. 37 and a subscript of 2 for the one in Eq. 54 to avoid confusion, we see that

$$\Pr\left[|\|\psi_P - \psi_Q\|_2 - \mathrm{MMD}(P, Q)| \geqslant \frac{11}{4}\varepsilon\right] \leqslant 2e^{-nc_1\varepsilon^2} + \mathcal{O}\left(\frac{1}{\varepsilon^2}\exp\left(\frac{-D\varepsilon^2}{64(d+2)}\right)\right) \tag{55}$$

$$+ 6e^{-\frac{n\varepsilon^2}{8}} + 8e^{-nc_2\varepsilon^2}. \tag{56}$$

This completes the proof and shows that embeddings $\psi^\pi$ used in practice enjoy the same asymptotic guarantees as $\phi^\pi$. The only difference is that the approximation error is increased by an additional term, which, crucially, also decreases exponentially as the number of samples $n$ increases.

To conclude, let us briefly compare the embeddings resulting from $\phi^\pi$ and $\psi^\pi$. Both are obtained by computing the empirical mean of RFF embeddings from $n$ independently sampled trajectories. The key difference lies in how they embed each trajectory: $\phi^\pi$ embeds a single state-action pair, sampled according to a geometric distribution, from each trajectory, whereas $\psi^\pi$ computes a weighted average of the embedding of all state action pairs within a trajectory. Both embeddings share the same expectation, but $\psi^\pi$ potentially exhibits lower variance because it averages the state-action embeddings over the entire trajectory. This benefit comes at the cost of an added approximation error term in the probabilistic bound obtained for $\psi^\pi$. However, this term is very similar to the other terms already present in the bound and, like them, decreases exponentially with $n$.

## C    CALIBRATED WEIGHTED PCA

Our calibrated and weighted PCA variant addresses three critical requirements for effective specification of behavioral descriptors in QD optimization: finding meaningful behavioral variations, ensuring compatibility with QD archives, and adapting to the evolving population of solutions. Below, we detail each component and its motivation.

Given policies $\{\pi_1, \ldots, \pi_m\}$ with embeddings $\Psi = [\psi^{\pi_1}, \ldots, \psi^{\pi_m}]^T$ and fitness scores (estimated returns) $\{f_1, \ldots, f_m\}$ our algorithm proceeds as follows:

**Step 1: Score normalization.** We normalize fitness scores to form a weight distribution:

$$\tilde{f}_i = \max\left( \frac{f_i - \min_j f_j}{\max_j f_j - \min_j f_j}, \frac{1}{m} \right) \tag{57}$$

$$w_i = \frac{\tilde{f}_i}{\sum_j \tilde{f}_j} \tag{58}$$

where the weights sum to 1. The $\frac{1}{m}$ term in Eq. 57 ensures a minimum contribution from each policy.

*Motivation:* While all policies provide information about the behavioral space, high-performing policies represent more successful strategies that we want to emphasize when discovering diverse behaviors. Low-performing policies often exhibit undesirable behaviors that should have less influence on our descriptors. The normalization ensures all policies contribute at least minimally while prioritizing those with higher fitness.

**Step 2: Weighted PCA.** We compute:

$$\mu = \sum_{i=1}^{m} w_i \psi^{\pi_i} \quad \text{(weighted mean)} \tag{59}$$

$$\hat{\psi}^{\pi_i} = \psi^{\pi_i} - \mu \quad \text{(centered embeddings)} \tag{60}$$

$$\tilde{\psi}^{\pi_i} = \sqrt{w_i}\hat{\psi}^{\pi_i} \quad \text{(weighted centered embeddings)} \tag{61}$$

We perform SVD on the weighted centered embeddings to obtain the top $k$ principal components $\mathbf{P} \in \mathbb{R}^{D \times k}$.

*Motivation:* PCA offers several advantages for our context:

- It provides an affine transformation that preserves the geometry of the original embedding space, maintaining relative distances between policies up to scaling and translation.
- Unlike non-linear dimensionality reduction techniques, it doesn't introduce distortions that could misrepresent behavioral similarities.
- It requires no additional hyperparameters or iterative training procedures.
- The orthogonality of principal components ensures that each behavioral measure captures a distinct aspect of policy behavior.

- By weighting the PCA computation, we focus on capturing variations among high-performing policies.

**Step 3: Calibration.** We compute the 5th and 95th percentile quantiles of uncalibrated projections $\hat{\text{desc}}(\pi) = \mathbf{P}^T(\psi^\pi - \mu)$ along each dimension:

$$\mathbf{q}_{\text{low}} = \text{quantile}(\{\hat{\text{desc}}(\pi_i)\}, 0.05) \tag{62}$$

$$\mathbf{q}_{\text{high}} = \text{quantile}(\{\hat{\text{desc}}(\pi_i)\}, 0.95) \tag{63}$$

The final transformation maps $[\mathbf{q}_{\text{low}}, \mathbf{q}_{\text{high}}]$ to $[-1, 1]^k$:

$$\mathbf{s} = \frac{2}{\mathbf{q}_{\text{high}} - \mathbf{q}_{\text{low}}} \tag{64}$$

$$\mathbf{c} = -\mathbf{1} - \mathbf{s} \cdot \mathbf{q}_{\text{low}} \tag{65}$$

$$\mathbf{A} = \text{diag}(\mathbf{s})\mathbf{P}^T \tag{66}$$

$$\mathbf{b} = \mathbf{c} - \mathbf{A}\mu \tag{67}$$

(Operations in the first two lines are element-wise)

*Motivation:* Calibration addresses a practical challenge in QD optimization:

- PCA naturally produces dimensions with different scales based on variance, which would require dimension-specific archive bounds.

- Calibration standardizes all dimensions to a fixed range $[-1, 1]$, allowing the QD algorithm to use consistent archive bounds.

- This standardization enables more uniform coverage of the archive along each dimension, preventing the QD algorithm from disproportionately exploring directions with naturally higher variance.

- The 5th/95th percentile choice ensures that most solutions fall within the archive bounds.

Importantly, the calibration step preserves the affine nature of the transformation, combining the projection and scaling into a single linear operation $\mathbf{A}$ with offset $\mathbf{b}$. This results in a computationally efficient mapping that maintains the essential geometric properties of the embedding space. The final behavioral descriptor $\text{desc}(\pi) = \mathbf{A}\psi^\pi + \mathbf{b}$ adaptively identifies and scales the most significant behavioral dimensions, focusing on variations among high-performing policies while ensuring compatibility with fixed-bound QD archives.

We conclude this section by noting an important caveat regarding the weighting strategy in cwPCA. In principle, weighting is designed to emphasize the contribution of high-performing solutions, guiding exploration toward more promising regions of the behavior space. However, this mechanism can sometimes be counterproductive by placing excessive emphasis on behaviors that are useful but ultimately suboptimal. For example, in a robotics task, a simple stabilization strategy such as preventing the robot from falling represents an accessible local optimum. While stabilization is beneficial in the early stages, if more advanced locomotion patterns have not yet been discovered, the weighting mechanism in cwPCA may disproportionately highlight variations of this basic strategy. This can lead to the discovery of a diverse set of stabilization behaviors that remain confined to a narrow and suboptimal region of the behavior space. Although this issue is less likely to happen in high-dimensional behavior spaces, it can hinder the performance in constrained behavior spaces where early suboptimal variations may be amplified and prematurely lead the search into a local optimum.

To assess the extent of this effect, we conducted an ablation study in the Walker2d environment, comparing the performance of AutoQD with and without fitness weighting in PCA. Table 3 summarizes the results of this ablation. While weighting slightly improved the performance across all metrics, the difference were not statistically significant according to a double-sided Mann-Whitney U test (p-value $> 0.7$), suggesting that the impact of weighting may vary case by case, depending on the structure of the behavior space and difficulty of exploration.

Table 3: Comparison of AutoQD with and without weighting. Reported values are mean ± standard error over evaluations with eight different random seeds.

| Method | GT QD Score ($\times 10^4$) | Mean Objective | Vendi Score |
|---|---|---|---|
| AutoQD (with weighting) | $17.74 \pm 3.85$ | $1162.8 \pm 116.8$ | $8.35 \pm 4.14$ |
| AutoQD (w/o weighting) | $17.74 \pm 3.51$ | $1143.6 \pm 83.8$ | $7.67 \pm 3.32$ |

# D  DETAILS OF CMA-MAE

Here we provide the pseudocode for both the initialization of CMA-MAE (Algorithm 2) as well as its update step (Algorithm 3). These were abstracted as function calls in Algorithm 1 in the main paper for the sake of clarity.

In these pseudocodes, note that the internal parameters of CMA-ES include a Gaussian "search" distribution that are used to sample candidate policy parameters (Line 4 of Algorithm 3) and are updated through CMA-ES (Line 17 of Algorithm 3). For a more detailed exposition of CMA-MAE, we refer the reader to Fontaine and Nikolaidis (2023).

---

**Algorithm 2** `CMA_MAE_Init`

---

1: **function** CMA_MAE_INIT($k$)
2:     **Input:** Behavior space dimension $k$
3:     **Output:** Empty archive $\mathbb{A}$, Optimization state QDState
4:     Initialize CMA-ES internal parameters: CMA_ES_State
5:     Initialize an empty archive $\mathbb{A}$          ▷ Uniform grid over $[-1.2, 1.2]^k$
6:     **for all** cells $e$ in $\mathbb{A}$ **do**
7:         $t_e \leftarrow \text{min\_objective}$          ▷ Acceptance threshold
8:     **end for**
9:     QDState $\leftarrow$ (CMA_ES_State, $\{t_e\}_{e \in \mathbb{A}}$)
10:     **return** $(\mathbb{A}, \text{QDState})$
11: **end function**

---

---

**Algorithm 3** `CMA_MAE_Step`

---

1: **function** CMA_MAE_STEP($\mathbb{A}$, QDState, desc)
2:     **Required Hyperparameters:** learning rate $\alpha$, batch size $\lambda$
3:     **for** $i = 1, \ldots, \lambda$ **do**
4:         Sample candidate: $\theta_i \sim \mathcal{N}(\theta_{\text{QDState}}, \Sigma_{\text{QDState}})$
5:         trajectories $\leftarrow$ `collect_rollouts`($\theta_i$)
6:         $f \leftarrow$ `mean_return`(trajectories)          ▷ Fitness
7:         Compute $\psi$ according to Eq. 6 from trajectories
8:         BD $\leftarrow$ desc($\psi$)
9:         $e \leftarrow$ `calculate_cell`($\mathbb{A}$, BD)          ▷ Locate corresponding cell from the archive
10:         $\Delta_i \leftarrow f - t_e$          ▷ Improvement over the cell's threshold
11:         **if** $f > t_e$ **then**
12:             Replace the current occupant of cell $e$ in the archive $\mathbb{A}$ with $\theta_i$
13:             $t_e \leftarrow (1 - \alpha)t_e + \alpha f$
14:         **end if**
15:     **end for**
16:     Rank $\theta_i$ by $\Delta_i$
17:     Adapt CMA-ES parameters based on improvement rankings $\Delta_i$
18:     **return** updated $\mathcal{S}$
19: **end function**

---

# E    IMPLEMENTATION DETAILS AND HYPERPARAMETERS

## E.1    ENVIRONMENTS

We use the latest versions of the environments available in Gymnasium (Towers et al., 2024) in our experiments:

- BipedalWalker-v3,
- Ant-v5,
- HalfCheetah-v5,
- Hopper-v5,
- Swimmer-v5,
- Walker2d-v5.

## E.2    NETWORK ARCHITECTURE

All QD-based methods (AutoQD, Aurora, LSTM-Aurora, and RegularQD) use identical policy architectures: a neural network with two hidden layers of 128 units each and tanh activation functions. These networks employ a Toeplitz structure, which constrains the weight matrices such that all entries along each diagonal share the same value (Choromanski et al., 2018). This constraint enforces parameter sharing and reduces the search space.

SMERL uses a similar network architecture but with ReLU activations and without the Toeplitz constraint. Since SMERL employs gradient-based RL optimization, the Toeplitz structure is not necessary. We use ReLU activations to keep consistency with the author's hyperparameters and the open source implementations.

DvD-ES uses the authors' provided implementation, which employs MLPs with two hidden layers of size 32.

## E.3    QD ALGORITHM CONFIGURATION

All methods utilize the standard Pyribs (Tjanaka et al., 2023b) implementation of CMA-MAE (Fontaine and Nikolaidis, 2023) as the underlying QD algorithm. They employ grid archives that are discretized to 10 cells along each dimension and use 5 emitters with different initial step sizes of $\{0.01 \times 2^i\}_{i=1}^5$. The rest of the configuration is presented below.

Table 4: Common QD algorithm parameters shared across all methods

| Parameter | Value |
| --- | --- |
| Number of CMA-ES Instances | 5 |
| Initial Step Size ($\sigma_0$) | $\{0.01 \times 2^i\}_{i=1}^5$ |
| Batch Size | 64 |
| Restart Rule | 100 iterations |
| Archive Learning Rate | 0.01 |
| Total Iterations | 500 |
| Evaluations per Policy | 5 |

## E.4    AUTOQD

Our proposed method uses Random Fourier Features (RFF) to map trajectories/policies into embeddings and progressively refines a measure map during optimization using calibrated weighted PCA to convert policy embeddings into low-dimensional behavior descriptors. The embedding map normalizes the observations based on the trajectories that it observes throughout its lifetime.

The update schedule indicates the iterations at which the measure map is refined using the current archive.

Table 5: AutoQD-specific parameters

| Parameter | Value |
|---|---|
| *RFF Embedding* | |
| Embedding Dimension | 100 |
| State Normalization | True |
| Kernel Width | $\sqrt{\text{state dim} + \text{action dim}}$ |
| Discount Factor ($\gamma$) | 0.999 |
| *Measure Map* | |
| Measures Dimension | 4 |
| Update Schedule | $[20, 50, 100, 200, 300]$ |

## E.5 AURORA

Aurora learns a behavioral characterization using an autoencoder that reconstructs states.

Table 6: AURORA parameters

| Parameter | Value |
|---|---|
| *Encoder Architecture* | |
| Mapping | $\mathcal{S} \to \mathbb{R}^4$ |
| Hidden Layers | $[64, 32]$ |
| Latent Dimension | 4 |
| *Decoder Architecture* | |
| Mapping | $\mathbb{R}^4 \to \mathcal{S}$ |
| Hidden Layers | $[32, 64]$ |
| *AutoEncoder Training* | |
| Max Epochs | 50 |
| Learning Rate | 0.001 |
| Batch Size | 64 |
| Validation Split | 0.2 |
| Early Stopping Patience | 10 |
| Update Schedule | $[20, 50, 100, 200, 300]$ |

At each iteration, the autoencoder is trained for a maximum of 50 epochs on $80\%$ of all data. A validation loss is computed using the remaining $20\%$ and if it does not decrease for 10 consecutive epochs, the training can stop earlier.

## E.6 LSTM-AURORA

This variant of AURORA uses an LSTM-based architecture to encode full trajectories. The encoder maps sequences of states to hidden states. The last hidden state of a trajectory is mapped to a latent vector (the behavioral descriptor). The decoder maps this latent back to a hidden state vector and reconstructs the trajectory starting with this hidden state and using teacher forcing.

The trajectory sampling frequency of 10 means that every 10th state in a trajectory is used for encoding, following the authors' implementation.

## E.7 REGULARQD

The baseline RegularQD method uses handcrafted behavioral descriptors specific to each environment. For all of the environments except Swimmer, these are the foot-contact frequencies which are commonly used in literature. For Swimmer, we use three descriptors that measure angular span (i.e., how much the joints bend), phase coordination (i.e., how well the joints coordinate), and straightness (i.e., how straight the trajectory is).

Table 7: LSTM-Aurora parameters

| Parameter | Value |
|---|---|
| *Encoder-Decoder Architecture* | |
| Type | LSTM |
| Hidden Dimension | 32 |
| Latent Dimension | 4 |
| Hidden-to-Latent Map Type | Linear |
| Latent-to-Hidden Map Type | Linear |
| Teacher Forcing | True |
| Trajectory Sampling Frequency | 10 |
| *AutoEncoder Training* | |
| Epochs | 50 |
| Learning Rate | 0.001 |
| Batch Size | 64 |
| Validation Split | 0.2 |
| Early Stopping Patience | 10 |
| Update Schedule | $[20, 50, 100, 200, 300]$ |

### E.8    SMERL

Our implementation of SMERL is based on an open source implementation, modified slightly to make it compatible with the latest version of the environments and to add parallelization. Other than increasing the size of network's hidden layers (two hidden layers of size 128), doubling the number of skills to 10, and increasing the total training steps to $1.6 \times 10^7$ total timesteps, we keep the default hyperparameters.

### E.9    DvD-ES

We use the DvD-ES implementation provided by the authros (link) with slight modifications to make it compatible with the latest versions of the environments. Other than the number of policies (we use 10) we keep the default hyperparameters.

### E.10    EVALUATION

In Sec. 4.2 we stated that the Vendi Score (VS) relies on a positive-definite similarity kernel. For this purpose, we use the Gaussian (RBF) kernel, defined for a pair of embeddings $\mathbf{x}$ and $\mathbf{y}$ as:

$$K = \exp\left(-\gamma \|\mathbf{x} - \mathbf{y}\|^2\right) \tag{68}$$

To select an appropriate value of $\gamma$, we adopt a variant of the *median heuristic* (Garreau et al., 2017), which sets

$$\gamma = \frac{\ln 2}{\text{median}\left(\|\mathbf{x}_i - \mathbf{x}_j\|^2\right)}, \tag{69}$$

where the median is computed over the pairwise squared distances between all policy embeddings. This choice ensures that two embeddings separated by the median distance will have a similarity of $K = 0.5$, offering an intuitive scaling of the kernel. To reduce computational overhead for methods that generate a large number of policies, we randomly subsample up to 1000 embeddings when computing the median distance.

Finally, we emphasize that this similarity kernel is distinct from the one used to construct the random Fourier feature (RFF) embeddings described in Sec. 3.1.

### E.11    COMPUTATIONAL RESOURCES

All of our experiments were conducted on local machines with an *AMD Ryzen Threadripper PRO 5995WX 64-Cores* CPU, 64GB of memory and either an *NVIDIA GeForce RTX 4090* or *NVIDIA*

*GeForce RTX 3090* GPU. Each training run for any of the algorithms took $\leqslant 3$ hours except the experiments for the SMERL baseline which took around 1 day each. Furthermore, the evaluation of each population took $\leqslant 1$ hours.

# F    ADDITIONAL RESULTS FROM EXPERIMENTS

The results in the main paper primarily focus on aggregate measures of quality and diversity, namely GT QD Score and qVS. While these metrics capture the combined effect of quality and diversity, they do not reveal how each algorithm manages the inherent trade-off between the two. Achieving high diversity often comes at the expense of average quality, since generating novel behaviors requires deviating from the "optimal" behavior. For example, in bipedal locomotion, behaviors such as hopping on one leg or sliding forward are diverse but achieve lower quality compared to standard two-legged walking, as they result in slower forward motion.

To better understand this trade-off, we compare the normalized mean quality and diversity across all six evaluation tasks. Quality is measured as the mean fitness of all policies in a population, while diversity is measured by the Vendi Score. Both values are normalized to $[0, 1]$ per task to enable cross-task comparison (raw values are reported in Table 8). Figure 5 visualizes the trade-off as a scatter plot, with quality on the x-axis and diversity on the y-axis. Each point corresponds to the outcome of an algorithm on one task, averaged over three evaluation seeds. Ideally, populations would achieve both high quality and high diversity (top right corner). The diagonal line represents a balanced 1-to-1 trade-off and points above it indicate more efficient quality-diversity trade-offs.

Several observations can be drawn from Figure 5. First, AutoQD achieves 5/6 points above the diagonal, highlighting its effectiveness in balancing quality and diversity. The next-best method, RegularQD, achieves 4/6, while no other baseline exceeds 2/6. Second, AutoQD tends to sacrifice some quality to achieve higher diversity, often producing the most diverse populations. This suggests that its learned behavior descriptors are particularly effective at capturing diverse behavioral variations. By contrast, DvD-ES typically achieves the highest mean quality but shows very limited diversity, with points concentrated in the bottom right.

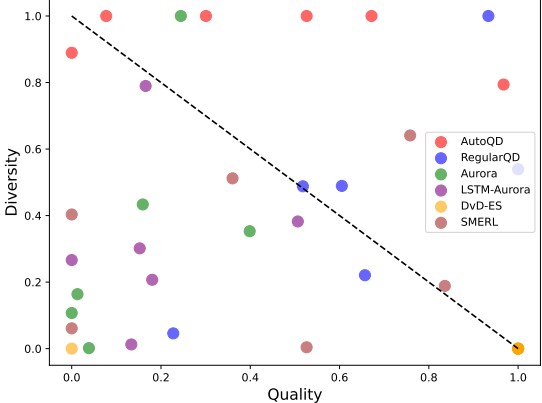

Figure 5: **Quality-diversity trade-off of algorithms across domains.** The x-axis shows normalized mean fitness (quality) and the y-axis shows normalized Vendi score (diversity). Each point corresponds to the outcome of an algorithm in one of the six domains. Points above the diagonal line exhibit a more favorable trade-off of quality for diversity.

**Fine-grained results.** Table 8 reports more detailed statistics for all methods: mean fitness, maximum fitness, Vendi Score, and coverage of the ground-truth archive (i.e., the number of occupied cells in an archive with hand-crafted descriptors). While the first two metrics indicate quality and the latter two indicate diversity, interpreting them in isolation can be misleading. For example, mean objective tends to favor smaller populations (e.g., DvD-ES, SMERL) or those with correlated behavior spaces (e.g., Aurora), as the fraction of high-performing policies in a large, independent archive

decreases exponentially (see ablations in Appendix H for more empirical evidence). Similarly, GT archive coverage is biased toward larger populations, since smaller populations may have too few policies to fill the descriptor space uniformly.

Even with these caveats, several patterns can be observed. First, SMERL, as an RL-based method, consistently achieves the highest maximum fitness in 5/6 tasks. This aligns with prior work (Tjanaka et al., 2022a), since reward functions in continuous-control tasks are often shaped for RL optimization, making them less suited to evolutionary strategies like CMA-MAE (Pagliuca et al., 2020). That said, advances in QD algorithms are rapidly narrowing this gap (Batra et al., 2024). Second, in HalfCheetah and Walker2d, the two domains where AutoQD underperforms compared to baselines, the causes differ. In **HalfCheetah**, AutoQD produces populations with significantly lower mean fitness. Rollout inspection revealed that many learned behaviors involved sliding close to the ground, propelled by small, rapid leg movements. Although these policies were diverse (as reflected in both Vendi Score and GT archive coverage), their quality remained too low to compete with the baselines. In **Walker2d**, the opposite occurred: AutoQD achieved high mean fitness but underperformed RegularQD in terms of diversity. Here, AutoQD tended to focus on variations of gaits dominated by the lower joints, while neglecting behaviors involving upper-body movement.

In both domains, the issue appears to be linked with early convergence to specific behavioral modes. Sliding in HalfCheetah and lower-joint motions in Walker2d are highly stable and thus readily discovered, making them likely attractors early on during training. Given that we restrict the learned behavior space to a low-dimensional (4-d) subspace, it is plausible that only variants of these behaviors are captured, constraining further exploration. Future work may mitigate this by employing higher-dimensional descriptors or pruning the learned descriptors, for example via extinction events (Lehman and Miikkulainen, 2015).

The last column of Table 8 shows the performance of a gradient-based QD algorithm, PGA-ME (Nilsson and Cully, 2021), when using ground truth behavior descriptors, similar to the RegularQD baseline. The performance of this algorithm, in particular its ability to outperform RegularQD and its high maximum objective value, serves to illustrate the potential of replacing CMA-MAE with more powerful QD algorithms. As noted in the paper, combining AutoQD's learned descriptors with gradient-based QD methods such as PGA-ME is non-trivial. However, the results presented here indicate strong potential for the development of such algorithms. One primary area where AutoQD lagged behind some baselines is pure optimization capability, which manifested as lower maximum policy performance in some domains. PGA-ME's strong performance in this regard reinforces our hypothesis that this limitation can be addressed by replacing CMA-MAE with stronger QD algorithms, highlighting a viable direction for future work.

## G  ADDITIONAL RESULTS FROM ADAPTATION EXPERIMENTS

Here, we provide further data and analysis derived from our adaptation experiments, discussed in Section 4.4.

Figure 6 is analogous to Figure 4 presented in the main paper and illustrates the number of policies that successfully adapt to changes in the robot mass scale. It reveals a similar pattern in that AutoQD's population consistently includes more successful policies under the strict threshold of $p = 0.9$ and generally maintains the second-highest count (behind LSTM-Aurora) under the relaxed threshold of $p = 0.7$.

We also include the full distribution of policy returns under five different friction coefficients and mass scales in Figure 7 and Figure 8, respectively. To facilitate clearer comparison among the top-performing methods, we exclude RegularQD and SMERL from these figures, as they consistently exhibited poorer performance in this domain. These box plots show that AutoQD's population generally exhibits a lower mean return, suggesting that many of its policies fail to adapt to the altered environmental conditions. Crucially, however, it also exhibits **longer tails toward high returns**, indicating that it has discovered a number of policies that are exceptionally effective at adapting to the environmental changes. This contrasts with, for instance, DvD-ES, which primarily displays the inverse behavior. Policies found by DvD-ES maintain a relatively high mean performance but exhibit much less variance and shorter tails. Consequently, while DvD-ES generally achieves the

Table 8: Fine-grained results from the main experiments. Similar to the results presented in the main paper, the reported values are the mean $\pm$ standard error over evaluations with three different random seeds.

| Metric | AutoQD | RegularQD | Aurora | LSTM-Aurora | DvD-ES | SMERL | PGA-ME |
|---|---|---|---|---|---|---|---|
| **Ant** | | | | | | | |
| Mean Objective | $656.09 \pm 16.95$ | $988.09 \pm 5.24$ | $15.69 \pm 11.69$ | $112.27 \pm 88.69$ | $-23.01 \pm 3.59$ | $509.20 \pm 121.00$ | $1540.15 \pm 106.75$ |
| GT Coverage | $2046.33 \pm 4.84$ | $918.00 \pm 11.37$ | $40.00 \pm 12.06$ | $124.67 \pm 11.61$ | $3.00 \pm 1.00$ | $6.00 \pm 1.53$ | $185.00 \pm 4.16$ |
| Vendi Score | $72.37 \pm 10.63$ | $39.49 \pm 3.93$ | $1.11 \pm 0.01$ | $1.90 \pm 0.54$ | $1.00 \pm 0.00$ | $1.28 \pm 0.18$ | $8.95 \pm 1.50$ |
| Max Objective | $1719.21 \pm 73.15$ | $1422.68 \pm 46.59$ | $1440.65 \pm 226.10$ | $1398.91 \pm 202.09$ | $-9.00 \pm 4.35$ | $2481.71 \pm 638.22$ | $3885.86 \pm 661.89$ |
| **HalfCheetah** | | | | | | | |
| Mean Objective | $903.35 \pm 142.79$ | $2526.13 \pm 294.52$ | $1669.46 \pm 240.10$ | $1422.47 \pm 204.38$ | $4036.59 \pm 70.11$ | $2032.38 \pm 401.47$ | $3141.56 \pm 74.26$ |
| GT Coverage | $237.67 \pm 23.14$ | $94.00 \pm 2.65$ | $66.33 \pm 4.91$ | $78.33 \pm 5.36$ | $2.00 \pm 0.58$ | $6.33 \pm 1.45$ | $368.00 \pm 4.04$ |
| Vendi Score | $5.29 \pm 1.59$ | $3.44 \pm 0.34$ | $5.80 \pm 0.81$ | $4.83 \pm 0.16$ | $1.19 \pm 0.11$ | $3.55 \pm 0.56$ | $5.07 \pm 0.06$ |
| Max Objective | $3392.38 \pm 564.68$ | $4421.82 \pm 370.81$ | $4398.90 \pm 590.30$ | $3843.73 \pm 677.20$ | $4314.53 \pm 43.11$ | $6280.03 \pm 623.47$ | $7853.83 \pm 162.90$ |
| **Hopper** | | | | | | | |
| Mean Objective | $1002.51 \pm 29.31$ | $1105.31 \pm 32.79$ | $527.87 \pm 65.83$ | $321.75 \pm 30.94$ | $1615.54 \pm 418.83$ | $1302.92 \pm 229.27$ | $2254.31 \pm 23.40$ |
| GT Coverage | $12.33 \pm 0.67$ | $9.67 \pm 0.33$ | $18.33 \pm 0.33$ | $19.00 \pm 0.58$ | $3.33 \pm 0.67$ | $7.33 \pm 0.33$ | $18.33 \pm 0.33$ |
| Vendi Score | $4.50 \pm 0.20$ | $2.85 \pm 0.04$ | $2.67 \pm 0.09$ | $2.13 \pm 0.29$ | $1.27 \pm 0.13$ | $3.34 \pm 0.24$ | $3.23 \pm 0.22$ |
| Max Objective | $2018.32 \pm 478.94$ | $1992.65 \pm 92.00$ | $1234.09 \pm 37.85$ | $1344.32 \pm 129.88$ | $1816.53 \pm 534.66$ | $3087.78 \pm 92.50$ | $3768.64 \pm 47.62$ |
| **Swimmer** | | | | | | | |
| Mean Objective | $131.60 \pm 12.82$ | $240.70 \pm 5.06$ | $162.21 \pm 7.62$ | $194.67 \pm 14.52$ | $345.95 \pm 7.66$ | $39.61 \pm 1.41$ | $242.26 \pm 1.61$ |
| GT Coverage | $1324.67 \pm 186.63$ | $463.67 \pm 5.78$ | $446.67 \pm 41.83$ | $523.67 \pm 51.54$ | $6.33 \pm 0.67$ | $5.00 \pm 0.58$ | $571.00 \pm 6.43$ |
| Vendi Score | $16.92 \pm 3.68$ | $4.67 \pm 0.35$ | $6.75 \pm 0.25$ | $7.21 \pm 1.95$ | $1.20 \pm 0.13$ | $2.16 \pm 0.57$ | $5.94 \pm 0.43$ |
| Max Objective | $359.76 \pm 0.85$ | $349.12 \pm 1.33$ | $361.71 \pm 0.27$ | $359.60 \pm 0.96$ | $356.65 \pm 1.25$ | $43.62 \pm 0.65$ | $355.89 \pm 1.36$ |
| **Walker2d** | | | | | | | |
| Mean Objective | $1173.78 \pm 102.76$ | $1150.78 \pm 64.45$ | $519.84 \pm 40.85$ | $623.01 \pm 80.49$ | $1196.28 \pm 18.53$ | $1085.01 \pm 151.02$ | $1747.65 \pm 39.87$ |
| GT Coverage | $156.67 \pm 25.78$ | $91.33 \pm 0.33$ | $151.00 \pm 12.66$ | $215.67 \pm 8.88$ | $4.67 \pm 0.88$ | $9.67 \pm 0.33$ | $319.67 \pm 4.10$ |
| Vendi Score | $8.40 \pm 3.20$ | $10.17 \pm 0.89$ | $2.50 \pm 0.13$ | $4.17 \pm 0.47$ | $1.58 \pm 0.29$ | $3.20 \pm 0.17$ | $9.46 \pm 1.10$ |
| Max Objective | $2278.29 \pm 315.46$ | $2234.76 \pm 203.43$ | $1936.61 \pm 398.75$ | $2098.49 \pm 223.70$ | $1306.50 \pm 40.28$ | $3800.25 \pm 370.32$ | $4645.52 \pm 325.06$ |
| **BipedalWalker** | | | | | | | |
| Mean Objective | $-33.18 \pm 7.66$ | $2.30 \pm 0.98$ | $-47.79 \pm 4.41$ | $-8.74 \pm 18.30$ | $182.07 \pm 5.38$ | $-51.42 \pm 11.33$ | $105.50 \pm 11.81$ |
| GT Coverage | $332.33 \pm 4.48$ | $89.33 \pm 0.88$ | $202.67 \pm 12.91$ | $221.67 \pm 8.09$ | $2.33 \pm 0.88$ | $9.33 \pm 0.67$ | $240.67 \pm 8.84$ |
| Vendi Score | $12.17 \pm 0.52$ | $1.57 \pm 0.03$ | $2.88 \pm 0.21$ | $3.36 \pm 0.46$ | $1.06 \pm 0.00$ | $5.54 \pm 0.42$ | $4.17 \pm 0.06$ |
| Max Objective | $239.43 \pm 19.47$ | $6.95 \pm 1.33$ | $253.86 \pm 5.36$ | $250.54 \pm 11.63$ | $196.44 \pm 4.66$ | $190.52 \pm 46.35$ | $331.99 \pm 6.58$ |

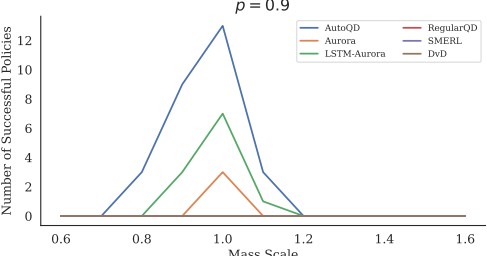 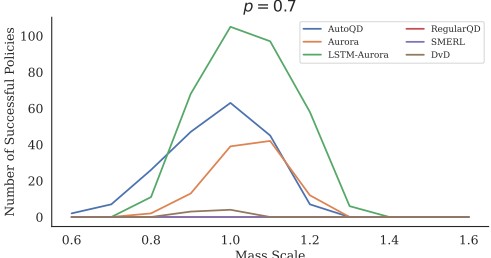

Figure 6: Number of successfully adapting policies in each population under changing mass. A policy is considered successful if its mean return is at least $Rp$, where $R$ is the highest overall return achieved in the unaltered environment. Results are shown for two success thresholds: $p = 0.9$ (left) and $p = 0.7$ (right).

highest mean return in Figures 7 and 8, it contains fewer policies that are close-to-best, as evidenced in Figures 4, 6, and 3.

We note that this distribution of returns is not unexpected: when a diverse collection of policies is generated, it is natural that many of them will perform poorly when substantial changes are introduced to the environment. However, the critical requirement is that a *subset* of these policies must maintain relatively high performance despite the changes in dynamics. Indeed, our expectation from a diverse population is not overall robustness across all policies; rather, we seek to include *some* policies that are *significantly* better than others at adapting to the given changes.

## H  EFFECT OF EMBEDDING AND BEHAVIOR SPACE DIMENSIONS

We perform two ablation studies to investigate how the dimensionality of (i) the behavior descriptors and (ii) the random Fourier feature embeddings affects performance. All experiments are conducted on the BipedalWalker environment using three different random seeds. We report the mean and standard error (shown as error bars) for three key metrics:

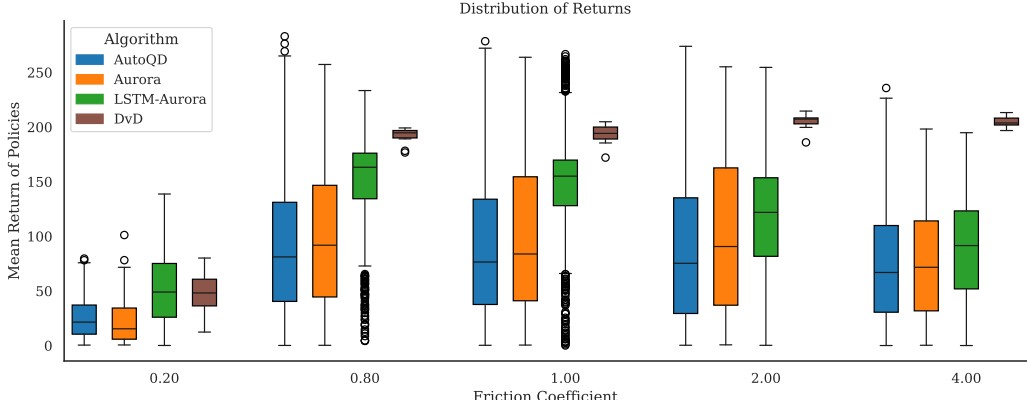

Figure 7: Distribution of policy returns for populations generated by each algorithm, evaluated under five different friction coefficients. Policies with negative returns are excluded.

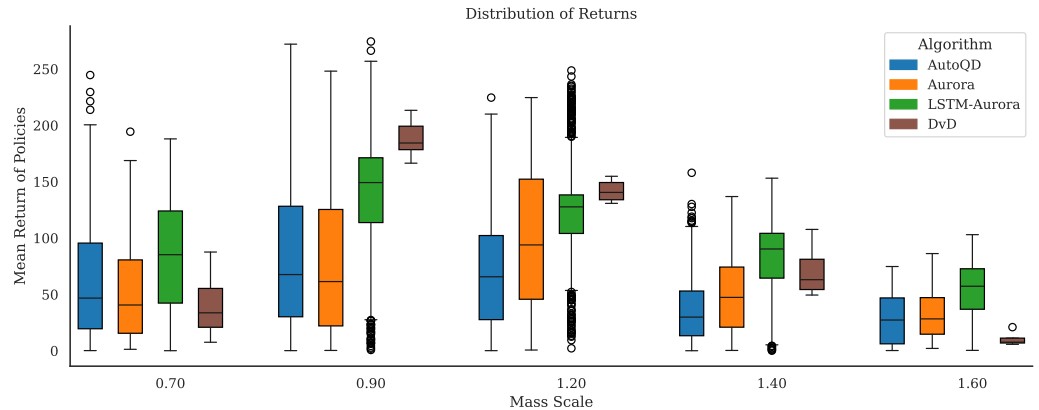

Figure 8: Distribution of policy returns for populations generated by each algorithm, evaluated under five different mass scales. Policies with negative returns are excluded.

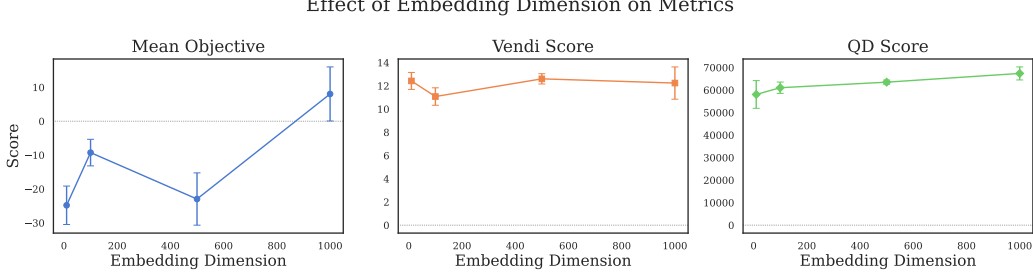

Figure 9: Ablating RFF embedding dimension on BipedalWalker. The plot report mean values over 3 random seeds with bars indicating standard errors.

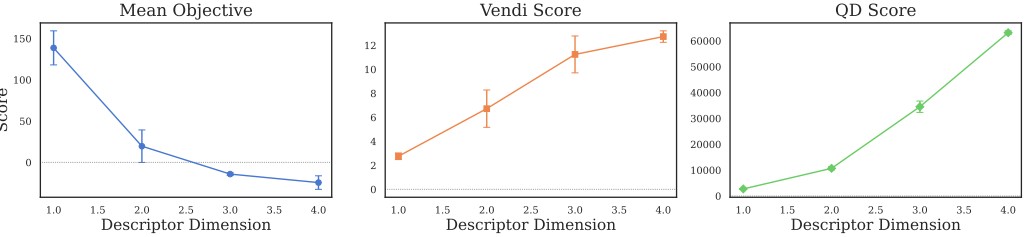

Figure 10: Ablating BD's dimension on BipedalWalker. The plot report mean values over 3 random seeds with bars indicating standard errors.

**Mean Objective.** The average of the objective (fitness) values across all policies found by the algorithm with a given configuration. It represents the overall quality of discovered solutions.

**Vendi Score.** A measure of behavioral diversity. Recall that it can be interpreted as the effective population size.

**QD Score.** The Ground Truth QD score introduced as in the main paper. It captures both the quality and the human-interpretable diversity of the discovered policies.

Note that quality-weighted Vendi Score (qVS) used in the main paper is just the product of the mean objective, normalized and scaled to be in $[0, 1]$, with the raw Vendi score. Here we report the raw Vendi score and the unnormalized mean objectives separately to provide a clearer picture of quality and diversity independently.

For the behavior descriptors, we vary the dimensionality of the measure space from 1 to 4. Fig. 10 presents the results which show a consistent improvement in both the QD score and the Vendi score as the dimensionality increases. In contrast, the mean objective value decreases with higher descriptor dimensionality. This suggests that as the behavior space becomes more complex, the fraction of high-quality solutions among all discovered solutions declines. This trend is intuitive: if we assume that a fixed proportion $\varepsilon \in (0, 1)$ of behaviors along each axis are high-performing, then the overall fraction of high-quality solutions decreases exponentially with the number of descriptor dimensions. Additionally, lower-dimensional behavior spaces may be easier for the underlying CMA-ES optimizer to search effectively for high-quality solutions.

For the embeddings, we vary the number of random Fourier features from 10 to 1000, evaluating configurations with 10, 100, 500, and 1000 dimensions. Fig 9 presents the results which suggest that performance is relatively robust to this hyperparameter. While our main experiments use 100-dimensional embeddings, even with as few as 10 features, AutoQD achieves competitive performance in terms of both quality and diversity.

## I   STABILITY OF LEARNED BEHAVIOR DESCRIPTORS

In this section, we examine the stability of the behavior descriptors (BDs) generated by AutoQD and the relevant baselines.

AutoQD, along with other baselines that rely on BDs (RegularQD, Aurora, and LSTM-Aurora), estimates the behavior of a policy by averaging the descriptor over multiple independently sampled trajectories. More formally, the estimated descriptor is calculated as:

$$\text{desc}(\pi) = \frac{1}{n} \sum_{i=1}^{n} \text{desc}(\tau_i),$$

where $\tau_1, \ldots, \tau_n$ are $n$ trajectories sampled from the policy $\pi$.

In our main experiments, we used $n = 5$ rollouts for all methods to estimate a given policy's BD. A descriptor function is considered "stable" if the variance of the descriptor output is low, meaning

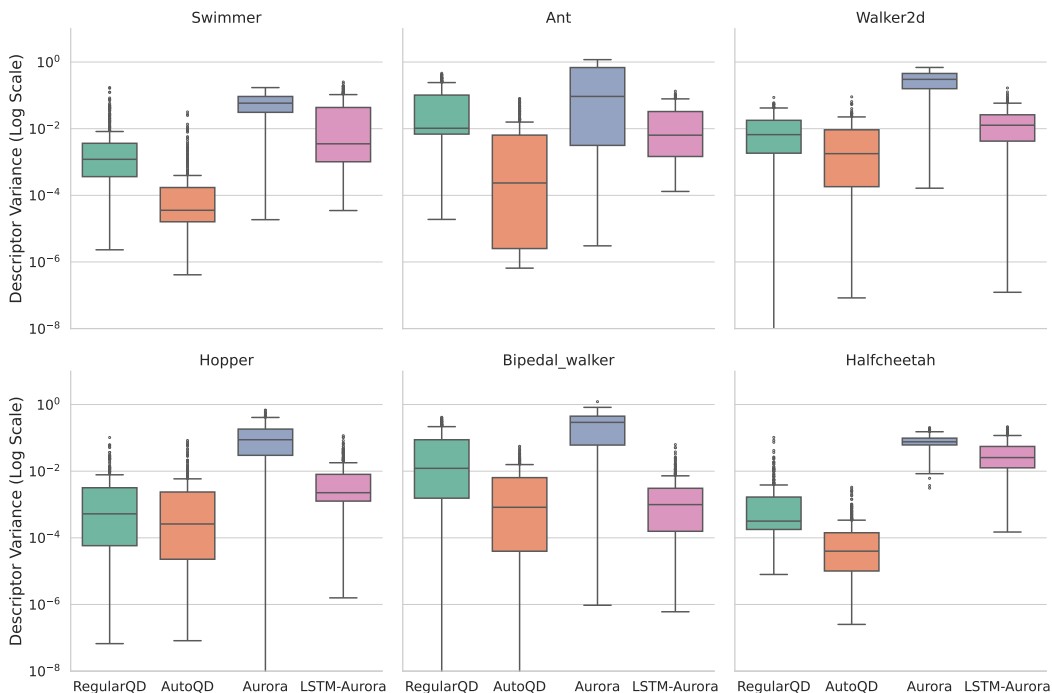

Figure 11: Distribution of variances of BDs assigned to 400 policies by different methods' descriptor function.

that different trajectories generated by the same policy are mapped to roughly the same behavior vector. This stability is desirable as it allows for an accurate policy behavior estimate using only a few sampled trajectories.

To investigate and compare the stability of the descriptor function learned by each method, we randomly sampled up to 100 policies from the final archives generated by AutoQD, RegularQD, Aurora, and LSTM-Aurora. We then collected 32 rollouts for each of these 400 policies. The trajectories were subsequently encoded as descriptors using the trained descriptor function of the four aforementioned algorithms. (For RegularQD, the descriptor function is hand-crafted rather than learned.) Since different BDs may inherently have different scales, we scaled the generated descriptors for each method to ensure they all lie within the range $[0, 1]$. Finally, we computed the variance (across the 32 rollouts) of each policy's embedding and plotted the distribution of these 400 variances.

Figure 11 illustrates these distributions across all six domains. The results demonstrate that AutoQD's descriptors are typically highly stable, achieving variances that are often orders of magnitude lower than those of the other methods in most environments. This finding suggests that AutoQD can reliably and with low variance encode a given policy's behavior based on just a few rollouts.

To further validate this point, we conducted an ablation study in the BipedalWalker environment. We compared the baseline version of AutoQD (which estimates descriptors using 5 rollouts) against two variants that use fewer ($n = 2$) or more ($n = 10$) rollouts for the estimate. As shown by the results in Table 9, increasing the number of rollouts to 10 yields only a marginal increase in performance, and the overall differences between these variants are relatively minor. Therefore, based on the experiments in this section, we can conclude that AutoQD's embeddings are robust, allowing for an accurate BD estimate using a small number of trajectories. This inherent stability enables AutoQD to maintain good performance even in constrained settings where as few as two trajectories are used to estimate the BDs.

Table 9: Comparison of AutoQD's performance when it uses different number of rollouts to estimate the behavior descriptors of policies. Reported values are mean $\pm$ standard error over evaluations with three different random seeds.

| Method | GT QD Score ($\times 10^4$) | Mean Objective | Vendi Score |
|---|---|---|---|
| AutoQD ($n = 2$) | $5.99 \pm 0.52$ | $-27.36 \pm 2.91$ | $10.42 \pm 0.61$ |
| AutoQD ($n = 5$) | $6.09 \pm 0.22$ | $-33.18 \pm 7.66$ | $12.17 \pm 0.52$ |
| AutoQD ($n = 10$) | $6.12 \pm 0.39$ | $-22.21 \pm 3.18$ | $12.46 \pm 1.01$ |

## J   QUALITATIVE ANALYSIS OF BEHAVIORS

In this section, we provide visual examples of the diverse behaviors discovered by AutoQD across different environments.

Since the learned behavior descriptors (BDs) generated by AutoQD do not necessarily align with simple, pre-defined behavioral variations (e.g., they may capture complex combinations of variations), we adopted the following procedure to select a representative sample of the variety present in the archive. First, we filtered the top 20% of best-performing policies from the final archive. We then started by selecting the single best-performing policy and iteratively selected the policy that was farthest (in AutoQD's learned behavior space) from the current set of selected policies, adding it to our selection pool. Using this procedure, we selected 10 diverse and high-performing policies from AutoQD's archive in each environment. We then collected rollouts from these policies and manually inspected them to choose those that exhibited the most noticeable differences.

Figures 12 and 13 showcase five distinct behaviors found in the Swimmer and BipedalWalker environments, respectively. Similarly, Figures 14 and 15 each showcase two behaviors from the HalfCheetah and Walker2d environments. A brief description of these behavioral variations is provided below.

In the **Swimmer** environment, we observed multiple distinct locomotion strategies, including: (1) Smooth forward movement using all joints in an S-shaped pattern. (2) Bending upwards to create a $\cup$-shaped pattern followed by a burst of upward motion. (3) Bending downwards to create a $\cap$-shaped pattern followed by a burst of forward motion. (4) Maintaining the body relatively straight while moving forward in a downward trajectory. (5) Moving forward by sharply bending the head and tail of the swimmer robot. Each row of Figure 12 depicts 8 sequential frames illustrating these five types of behavior.

From the **BipedalWalker** environment, we also selected five distinct behaviors: (1) Regular forward motion characterized by smooth hopping. (2) Extending the front leg to take large steps forward. (3) Sitting primarily on the back leg and crawling forward via small movements of the front leg. (4) Sitting on the back leg and moving forward through sharper, more aggressive movements of the front leg. (5) Standing tall and moving forward with minimal bending of the legs. These five behaviors are depicted sequentially in Figure 13.

We also include behaviors from the two environments where AutoQD did not achieve the highest performance. In **HalfCheetah**, we showcase two behaviors where: (1) the cheetah moves forward by taking large steps using the tip of its front leg, and (2) the head of the cheetah tilts slightly downwards such that a sizable part of the front leg makes contact with the ground during stepping. These two behaviors are presented in Figure 14. Overall, in this environment, the primary behavioral variation observed revolves around the frequency and speed of joint movements, rather than significant variations in posture.

Lastly, Figure 15 showcases two types of behaviors from **Walker2d**: (1) the robot keeps its legs close to one another and moves forward by "tip-toeing", and (2) the robot opens its legs widely and moves forward primarily through movements of the bottom joints. As noted in the paper, most policies in this environment rely heavily on movements from the bottom joints. While we do observe noticeable differences in posture (such as having the legs open or closed, as shown in the two examples), none of the policies appear to have learned to walk forward by taking large steps that coordinate the upper and bottom joints.

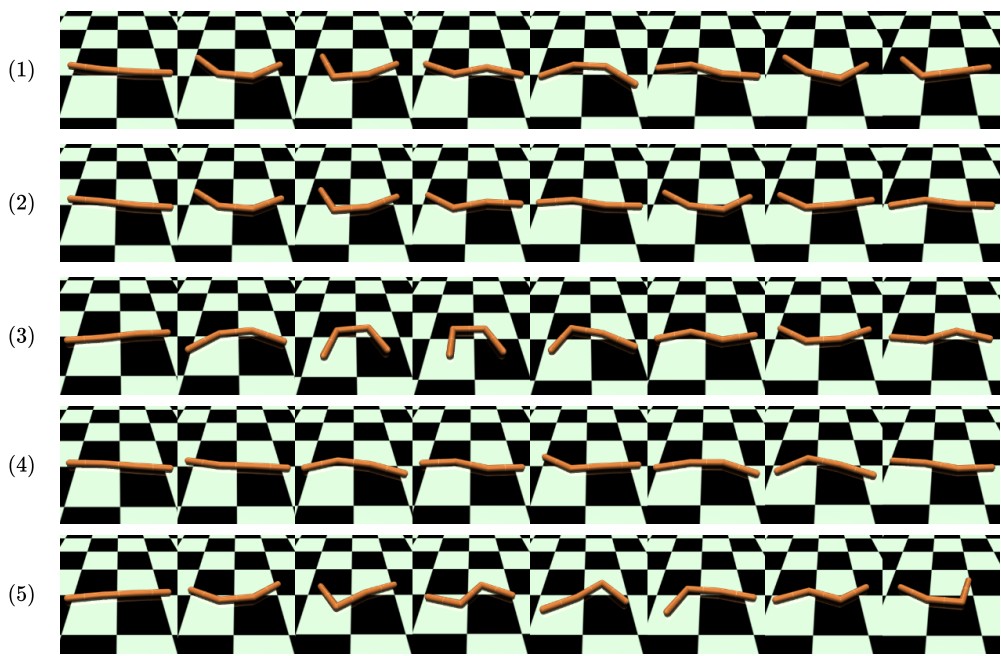

Figure 12: Demonstrating five different behaviors found by AutoQD in the Swimmer environment.

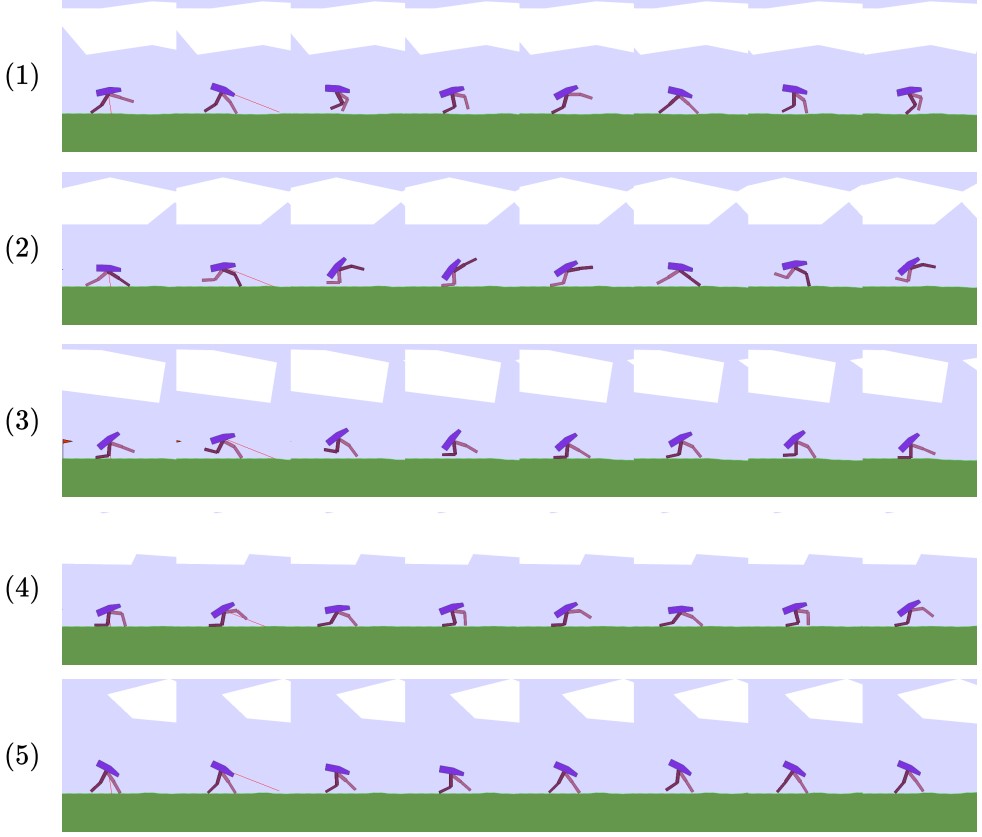

Figure 13: Demonstrating five different behaviors found by AutoQD in the BipedalWalker environment.

(1)
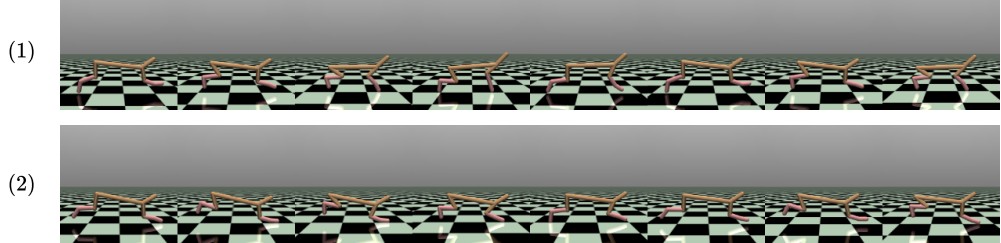

(2)

Figure 14: Demonstrating two types of behaviors found by AutoQD in the HalfCheetah environment.

(1)
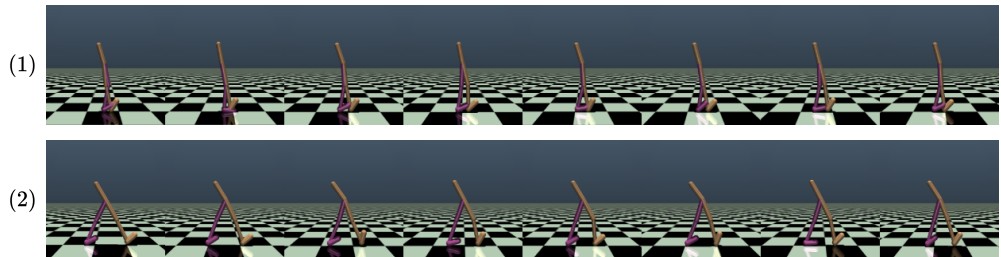

(2)

Figure 15: Demonstrating two types of behaviors found by AutoQD in the Walker2d environment.

## K    STATEMENT ON GENERATIVE AI USAGE

Generative AI tools were used as an aid to improve clarity and style in the writing of this paper.

