# OpenReview forum: "AutoQD: Automatic Discovery of Diverse Behaviors with Quality-Diversity Optimization"
_ICLR.cc/2026/Conference — ICLR 2026 Poster_

### Official Review · Reviewer_sHcp · 2025-10-24

**Soundness:** 2
**Presentation:** 2
**Contribution:** 2
**Rating:** 4
**Confidence:** 2

**Summary:**

This paper introduces AutoQD, a new method that solves a major problem in Quality-Diversity (QD) optimization: the need for hand-crafted behavioral descriptors (BDs). Instead of simple proxies, it represents a policy's entire behavior using its occupancy measure and uses Random Fourier Features to create a high-dimensional embedding of this occupancy measure. By projecting these embeddings into a low-dimensional space, AutoQD provides meaningful BDs to an "off-the-shelf" QD algorithm. This enables the unsupervised discovery of diverse, high-performing policies in reinforcement learning tasks without needing any predefined, domain-specific knowledge.

**Strengths:**

- It addresses a widely acknowledged and significant bottleneck in QD optimization: the reliance on hand-crafted, domain-specific BDs.
- The combination of the occupancy measures concept with Random Fourier Features to efficiently approximate the Maximum Mean Discrepancy is creative.

**Weaknesses:**

- In complex, high-dimensional state-action spaces (like a humanoid robot), a very large number of samples might be needed to get a stable and accurate embedding, potentially making the method computationally expensive or sample-inefficient.
- A key advantage of hand-crafted BDs is their human-interpretability (e.g., "walking speed," "jump height"). The BDs discovered by AutoQD are principal components of high-dimensional RFF embeddings, likely mathematically abstract and lacking any obvious physical meaning. This makes it difficult for researchers to understand what new behaviors are being discovered, which is often a primary goal of using QD.

Authors acknowledge these limitations.

**Questions:**

- Theorem 1 is based on i.i.d. samples, whereas the practical estimator ($\psi^{\pi}$) uses discounted, correlated samples from trajectories. Could you elaborate on how the theoretical bounds from Theorem 1 apply to this practical estimator, especially given the non-i.i.d. nature of the data?
- Regarding the periodic updates of the BDs, could you clarify the mechanism for managing the archive? Since updating the cwPCA projection effectively changes the coordinate system of the behavior space, how are existing policies in the archive handled? Is there a re-mapping process, and if so, how does the algorithm manage the computational cost and potential instability or 'forgetting' of previously discovered niches?
- The paper notes that AutoQD inherits the scalability challenges of the QD optimizer, but it also seems to introduce new computational steps, like storing all high-dimensional embeddings and periodically running PCA on the full archive matrix. Could you comment on the scalability of this new step as the archive size grows, and how significant this overhead is compared to the baseline QD algorithm?
- In the baseline comparisons, AutoQD uses a 4-dimensional BD space, while many of the RegularQD baselines use 2-dimensional descriptors. Given the paper's ablation study showing that performance scales with descriptor dimensionality, how can we disentangle the contribution of the AutoQD method itself from the contribution of simply using a higher-dimensional behavior space?
AutoQD's Dimension: Table 5 (Line 1029 on Page 20) explicitly states that AutoQD used a "Measures Dimension" of 4 for the main experiments.
Baseline's Implied Dimension: Section D.7 (Line 1070 on Page 20) describes the handcrafted descriptors for RegularQD. For robots like Walker2d, Bipedal Walker, and Hopper, it uses "foot-contact frequencies." These robots have two feet, which implies a 2-dimensional descriptor (one for each foot).

---

> ### Author Response · Authors · 2025-11-19
> **Part 1/2**
>
> We are thankful for your time and help in reviewing our paper. We are glad to hear that you found our work to be creative and addressing a widely acknowledged and significant bottleneck in QD optimization. We have revised the manuscript based on your feedback. Below, we address the concerns and questions that you raised.
>
>
> > [...] Could you elaborate on how the theoretical bounds from Theorem 1 apply to this practical estimator? [...]
>
> > [...] a very large number of samples might be needed to get a stable and accurate embedding [...]
>
> You are correct in that our algorithm uses $\psi^\pi$ vectors whereas the theoretical analysis considers $\phi^\pi$ as the policy embedding. In the revised version, we added Appendix B which extends our theoretical analysis to bridge this gap and we also discussed this point in Section 3.1. In summary, the fact that $\mathbb{E}[\phi^\pi] = \mathbb{E}[\psi^\pi]$ allows us to transfer the MMD-based guarantees to the $\psi^\pi$ embeddings. Specifically, we show that the distance between the empirical embeddings ($\psi^\pi$) of two policies concentrates around the MMD distance between their occupancy measures, with an additional error term that decays exponentially in $n$. The resulting bound (Eq. 53 in Appendix B) closely parallels the bound in Theorem 1 and retains the same asymptotic behavior.
>
> Additionally, we added more ablations in Appendix I (added in the revised version), to practically evaluate the stability of the behavior descriptors learned by AutoQD. In these experiments, we compare the stability (i.e., variance) of AutoQD's BDs with those of the baselines. Across all domains, AutoQD's descriptors are comparatively robust, often exhibiting variance *orders of magnitude* smaller than that of other methods.
>
> > Regarding the periodic updates of the BDs, could you clarify the mechanism for managing the archive? [...]
>
> You are correct in that the periodic updates of BDs essentially amount to a change of coordinates using the updated affine map of cwPCA. In practice, we store the full RFF embedding vector of each policy in the archive alongside the parameters of the policy. This allows us to quickly find the new BDs of any policy in the archive by applying the updated cwPCA map to the stored embedding without needing to collect new rollouts. The overhead cost of storing these embeddings is relatively low: In our experiments we used 100-dimensional embeddings (in our ablations we tested up to 1000-dimensional ones as well) whereas the policies typically have thousands of parameters. Therefore, the memory footprint of storing the embeddings is relatively low. That being said, in cases where memory is an issue, one can recompute the embeddings from newly sampled trajectories whenever they are needed.
>
> Regarding the second half of your question, since the cwPCA map is a simple affine transformation, operations involving it were stable and we did not face any instabilities using them. Furthermore, you are correct that after updating the cwPCA map and recomputing the BDs of solutions in the archive, the size of the archive decreases as some solutions will be mapped to the same cell and only the best one will be kept. While this does mean that certain behavioral niches may be "forgotten", we believe that this is both necessary and useful. It is necessary as keeping all of the policies that the algorithm encounters throughout its optimization process would lead to an archive that grows unbounded in size. Furthermore, it is useful because it allows new and more complex behavioral variations to replace older ones while keeping as much of the diversity as possible intact. For example, consider the walker environment. Initially, the policies struggle to maintain balance and the behavior descriptors may reflect different ways that the walker may fall down. As more and more stable policies are discovered, the updated behavior space will focus more on these successful behaviors (due to the weighting in cwPCA) and the new behavior descriptors may now capture the different ways that a policy can balance on its legs. And so on. Given that we have a fixed sized archive, it is necessary to "forget" older, less useful variations like the different ways of falling down so that the archive (and the new BDs) has enough capacity to capture the more complex, and more useful variations of maintaining balance.

---

> ### Author Response · Authors · 2025-11-19
> **Part 2/2**
>
> > [...] Could you comment on the scalability of this new step as the archive size grows, and how significant this overhead is compared to the baseline QD algorithm?
>
> AutoQD adds a computational overhead to the traditional QD pipeline in two ways.
> First, as you mentioned, is the cost that we incur for computing the PCA of the policy embeddings for all policies in the archive. In general, for an archive with $N$ policies and with $D$-dimensional behavior embeddings, PCA's time complexity is $\mathcal{O}(ND^2)$. Since we only need the top $d$ principle components (since our behavior space is $d$ dimensional) this complexity can be reduced to $\mathcal{O}(NDd)$. From a practical point of view, our archives have a maximum capacity of $N=10000$ (and they rarely have more than $N=5000$ policies stored at any given time), our RFF embeddings are $D=100$ dimensional, and the behavior space is $d=4$ dimensional. We also parallelized this computation on the GPU, which helps speed up the computation even further. Our training logs indicate that the wall-clock training time is by far dominated by rollout collection (about 8-10 seconds per iteration) and the periodic computation of the cwPCA takes less than 0.1 second.
> A second added overhead is the embedding of trajectories using the RFF map which happens every iteration. This includes computing a weighted sum over all collected state action pairs after a rollout collection step. This computation also leverages GPU parallelization and is therefore relatively fast. Our logs show that it takes between 0.4 to 1.2 seconds every iteration.
>
> Overall, the main computational overhead that AutoQD adds to a regular QD loop is the per-iteration trajectory embedding which increases the runtime by at most ~10%.
>
> > [...] how can we disentangle the contribution of the AutoQD method itself from the contribution of simply using a higher-dimensional behavior space? [...]
>
> Your observation about the dimension of the behavior space used by the RegularQD baseline is correct. Regular QD uses 1 dimensional embeddings for Hopper, 2 dimensional BDs for BipedalWalker, Walker2d, and HalfCheetah, 3 dimensional embeddings for Swimmer, and 4 dimensional embeddings for Ant.
>
> We chose these BDs for RegularQD because they are used extensively in prior work as BDs. Therefore, they provide an accurate representation of the hand-crafted descriptors that people use in practice and that we aim to compare with.
> Furthermore, being able to use larger behavior spaces is indeed an advantage of unsupervised QD methods compared to RegularQD which requires the manual design and implementation of each BD. So, it would be expected that automatically discovered BDs are in general higher dimensional compared to the set of hand-crafted BDs that a traditional QD method uses.
>
> As to your point about discerning the contribution of AutoQD's method from the effect of the behavior space size, we would like to point out the following two observations:
> 1. In the Ant domain, both AutoQD and RegularQD use 4 dimensional behavior spaces. Still, we observe that AutoQD outperforms RegularQD by a significant margin.
> 2. The two other unsupervised QD methods, Aurora and LSTM-Aurora, also use 4 dimensional behavior spaces. The fact that AutoQD outperforms them in most domains shows that AutoQD's good performance is not only due to having a larger behavior space, as those baselines operate under the same conditions as well.
>
> These two observations suggest that AutoQD's performance is not solely due to the fact that it uses a larger behavior space compared to the baselines.
>
>
> Thank you again for taking the time to review the paper and providing helpful feedback. We hope that these explanations and revisions sufficiently address your concerns. Please let us know if any further clarification or modification is required to satisfy your remaining concerns.

---

### Official Review · Reviewer_cGSg · 2025-10-30

**Soundness:** 2
**Presentation:** 3
**Contribution:** 2
**Rating:** 4
**Confidence:** 3

**Summary:**

This paper addresses the issue that Quality-Diversity (QD) optimization algorithms rely on hand-crafted Behavioral Descriptors (BDs) which limit exploration diversity, and proposes the AutoQD method. AutoQD leverages the one-to-one correspondence between policies and occupancy measures in MDPs, uses Random Fourier Features to approximate the Maximum Mean Discrepancy between occupancy measures for policy embedding, reduces high-dimensional embeddings to low-dimensional BDs via Calibrated Weighted PCA (cwPCA), and combines with the QD algorithm CMA-MAE to alternate between policy optimization and BD update. Theoretically, it proves that embeddings converge to the true MMD as the number of trajectories and embedding dimensions increase. Experimentally, on six continuous control tasks from the Gym library, AutoQD outperforms five baselines in most tasks in terms of GT QD Score, diversity, and adaptability to environmental changes (friction/mass variations), enabling the automatic discovery of diverse and high-performance policies without requiring domain knowledge.

**Strengths:**

1. The paper is easy to follow.

2. Solid theoretical support; Eliminates reliance on hand-crafted BDs; Compatible with existing QD algorithms and verified on mainstream tasks.

**Weaknesses:**

1. Low sample efficiency, needing many trajectories in stochastic environments.

2. Low-dimensional BDs may miss complex behaviors (e.g., ignoring leg-lifting in Walker2d).

3. Inferior maximum fitness compared to RL methods.

4. Fixed kernel bandwidth; poor scalability with large policy networks.

5. (Minor but Worth Discussing) A minor yet notable point worth discussing is the subjectivity of the concept of "diversity" itself. Diversity in QD optimization is inherently user-defined, as different users may prioritize distinct dimensions of behavioral variation for the same task. For example, in robot locomotion tasks, one user might focus on gait types (walking, hopping, sliding) as the core of diversity, while another could emphasize energy efficiency or adaptability to uneven terrain. However, AutoQD’s approach, which infers behavioral diversity from occupancy measure embeddings and MMD-based distances, essentially discovers implicit diversity definitions derived from data and mathematical metrics, rather than aligning with explicit user preferences. While this avoids the need for domain knowledge in hand-crafting BDs, it also raises the risk that the "diverse" policies AutoQD discovers may not fully match the specific diversity needs of practical users (e.g., a user caring about "load-carrying capacity-related behaviors" might find AutoQD’s focus on "trajectory smoothness" less relevant).

**Questions:**

1. How to avoid missing complex behaviors beyond increasing BD dimensions?

2. How to control computational complexity in high-dimensional observation tasks (e.g., images in Atari)?

3. What’s the path to combine with gradient-based QD for better sample efficiency?

---

> ### Author Response · Authors · 2025-11-19
> **Part 1/2**
>
> We are thankful for your time and help in reviewing our paper. We are glad to hear that you found our work to be theoretically sound and easy to follow. We have revised the manuscript based on your feedback. Below we address some of the concerns that you raised and answer your questions.
>
> > Low sample efficiency, needing many trajectories in stochastic environments.
>
> To quantitatively assess the number of trajectories that are needed for an accurate estimation of the behavior descriptors, we conducted some further experiments in Appendix I (added in the revised version).
> In it, we evaluated three choices of $n$ (number of trajectories used to estimate the descriptor) and observed that AutoQD is relatively robust to this parameter. Notably, using as few as $n=2$ trajectories does not substantially degrade performance.
> These findings are also consistent with the second set of experiments that we added in Appendix I, which compare the stability (i.e., variance) of AutoQD's behavior descriptors with those of the baselines. Across all domains, AutoQD's descriptors are comparatively robust, often exhibiting variance *orders of magnitude* smaller than that of other methods.
>
> Together, these experiments suggest that AutoQD is capable of accurately estimating descriptors from very few trajectories and that, in practice, it maintains its performance as we decrease its number of sampled trajectories.
>
> > [...] the subjectivity of the concept of "diversity" itself. [...]
>
> Traditionally, QD methods offer substantial flexibility in defining behavior descriptors. When users know exactly what kind of diversity they want, or when domain experts can specify suitable attributes, QD algorithms provide fine-grained control by allowing behavior descriptors to be defined manually. In other cases, users may not be able to articulate their desired notion of diversity, but we can still extract their preferences through comparative queries. Recent work on QDHF [1], which we cite in our paper, investigates how QD algorithms can operate in such settings and proposes methods for inferring human notions of diversity from comparative data.
>
> Our work fits into a third class of QD methods that assume a completely unsupervised setting, where no user-defined prior over diversity is available. In this case, we argue that occupancy measures provide a principled way to model policy behavior, and that exploring the resulting embedding space yields meaningfully diverse policies. This claim is supported quantitatively by AutoQD's high GT QD Score (main results, Table 1) and GT Coverage (Appendix F, Table 8). The ground-truth (GT) archive is defined using human-specified notions of diversity, so achieving high QD and Coverage scores indicates that the behaviors discovered by AutoQD form a superset of those that human designers judge to be diverse. In addition, we added a qualitative assessment in Appendix J which shows that AutoQD discovers behaviors that human users are also likely to find diverse and interesting.
>
> [1] Ding, Li, et al. "Quality Diversity through Human Feedback: Towards Open-Ended Diversity-Driven Optimization." _Forty-first International Conference on Machine Learning_.
>
> > How to avoid missing complex behaviors beyond increasing BD dimensions?
>
> While our current instantiation of AutoQD can already discover a wide range of behaviors, we see two main avenues for enabling it to uncover even more diverse and complex ones. The first is integrating AutoQD with state-of-the-art gradient-based QD methods such as PGA-MAP-Elites [1] and PPGA [2]. By leveraging modern RL machinery, which is known to be effective at producing complex behaviors, AutoQD could more easily explore intricate regions of the behavior space. We discuss the challenges associated with this integration in our response to your final question.
>
> The second avenue, as you noted, is increasing the dimensionality of the behavior space. Although AutoQD is robust to its hyperparameters in our ablations, higher-dimensional BDs consistently lead to performance improvements. Early in optimization, the algorithm naturally favors discovering simpler behavioral variations. A larger BD space provides more directions for exploration, enabling AutoQD to discover more complex behaviors.
> However, we must note that increasing BD dimensionality is non-trivial: most current QD algorithms (including CMA-MAE) do not scale gracefully with higher-dimensional descriptor spaces. Thus, progress in this direction requires improvements not only to AutoQD but also to the underlying QD optimization algorithms themselves.
>
> [1] Nilsson, Olle, and Antoine Cully. "Policy gradient assisted map-elites." _Proceedings of the Genetic and Evolutionary Computation Conference_. 2021.
> [2] Batra, Sumeet, et al. "Proximal Policy Gradient Arborescence for Quality Diversity Reinforcement Learning." _The Twelfth International Conference on Learning Representations_.

---

> ### Author Response · Authors · 2025-11-19
> **Part 2/2**
>
> > How to control computational complexity in high-dimensional observation tasks (e.g., images in Atari)?
>
> Our theoretical results suggest that the size of the RFF embeddings ($D$) should increase at least linearly with the size of the observation vectors ($d$) to maintain a fixed approximation error. Therefore, in high-dimensional, image-based observation spaces we may require very large RFF embeddings which could introduce additional challenges. A practical strategy in such settings would be to first encode images using a pre-trained or online-trained vision model and apply RFFs in the resulting latent space. While a full investigation is beyond the scope of this paper, we view AutoQD as a first step upon which such extensions can be built.
>
> > What’s the path to combine with gradient-based QD for better sample efficiency?
>
> *In principle*, AutoQD provides a general recipe for iteratively learning and refining the behavior space used by a QD algorithm. However, integrating this iterative refinement with gradient-based QD methods such as PGA-MAP-Elites [1] and PPGA [2] is non-trivial. These methods have more complicated training procedures which involve training critic networks with RL objectives that are known to be sensitive to non-stationary optimization targets. Because AutoQD dynamically updates the behavior space, the associated learning signals for these critics also change over time, which can destabilize training.
> Overall, we believe that there are two main challenges in integrating AutoQD with gradient-based methods. On a technical level, one would need to address the instability in the training of the critic, through regularization techniques or by adapting AutoQD's iterative method to encourage more smooth and gradual transformations of the behavior space instead of hard-updates in fixed intervals. On a practical level, one would also need to take extra care in selecting appropriate hyperparameters, as RL-based methods typically have many more hyperparameters and are more sensitive to their values.
> We have updated the future works section to specifically mention these challenges.
>
> [1] Nilsson, Olle, and Antoine Cully. "Policy gradient assisted map-elites." _Proceedings of the Genetic and Evolutionary Computation Conference_. 2021.
> [2] Batra, Sumeet, et al. "Proximal Policy Gradient Arborescence for Quality Diversity Reinforcement Learning." _The Twelfth International Conference on Learning Representations_.
>
>
> Thank you again for taking the time to review the paper and providing helpful feedback. We hope that these explanations and revisions sufficiently address your concerns. Please let us know if any further clarification or modification is required to satisfy your remaining concerns.

---

### Official Review · Reviewer_aLN2 · 2025-10-30

**Soundness:** 2
**Presentation:** 2
**Contribution:** 3
**Rating:** 4
**Confidence:** 3

**Summary:**

The paper introduces AutoQD, an algorithm for discovering diverse, high-performing policies without hand-crafted behavioral descriptors. The method combines a Quality-Diversity (QD) optimizer with an automated way to learn behavioral descriptors of policies.

AutoQD represents each policy through its occupancy measure (the distribution over state-action pairs the policy visits). To compare the behavior of two different policies, one can measure the distance between their occupancy measures using the Maximum Mean Discrepancy (MMD). Computing MMD directly is costly, so the authors approximate it using random Fourier features, which transform each occupancy measure (and hence each policy) into a finite-dimensional behavioral embedding. In this embedding space, Euclidean distances approximate the true MMD distances between policies (Theorem 1). Because QD optimizers operate over low-dimensional behavior spaces, AutoQD applies Calibrated Weighted PCA to these embeddings (weighted by their performance) to project them into compact behavioral descriptors. These descriptors and their performances are then passed into a standard QD algorithm (CMA-MAE) to discover a diverse set of high-performing policies (see Figure 1 for an illustration of the entire pipeline).

**Strengths:**

[Originality] Proposes a new theoretically motivated connection between occupancy measures and QD behavior descriptors, moving beyond handwritten heuristics.

[Quality] Method is modular and seems to be somewhat compatible with different QD optimization methods (authors only test CMA-MAE QD methods, and I have a question regarding gradient-based methods)

[Quality] Implementation details are well-documented in Appendix D. The code is well structured (I haven't run it, but I have read some of it and was able to find the main parts of the algorithm).

[Clarity] Introduction and motivation are clear and well-written.

[Clarity] Figure 1 and Algorithm 1 make the method easy to understand. I would add some "labels" to the different components of Figure 1 (e.g. QD archive, behavioral embeddings, etc).

[Significance] Addresses a limitation of QD methods - the need for hand-crafted behavior descriptors in a principled way.

**Weaknesses:**

[Method]

- Computing accurate embeddings requires many trajectories, which might be very sample inefficient in long-horizon or high-variance tasks.

- The kernel bandwidth and embedding dimension are fixed globally, but they could have a significant influence on performance.

[Experiments]


- Narrow experimental scope.

The experiments are mainly against QD methods that also use CMA-MAE (The experiments also include another evolutionary method (DvD-ES) and an RL method (SMERL) which are not very competitive in the tested domains). To clarify, there are no gradient-based QD baselines (e.g. DQD, PGA-MAP-Elites, PPGA). If AutoQD should be seen as an improvement to other CMA-MAE QD methods, then the paper should frame AutoQD as a CMA-MAE enhancement. Otherwise, the paper needs more broad experimental validation to support a claim of superiority.

- Lack of failure analysis. (Minor)

The authors briefly acknowledge underperformance of AutoQD on Walker2d and HalfCheetah (section 4.3), but don't have any deep analysis or visual evidence. Exploring failure cases more systematically would strengthen the paper, e.g., by visualizing the learned behaviors or the structure of the descriptor space to provide intuition about what AutoQD fails to capture. Generally, it's unclear what the gained diversity looks like (see later point).

- Misalignment between motivation and evaluation.

The introduction motivates quality-diversity through robustness and open-ended learning (i.e., diverse policies should help under dynamic shifts or enable new capabilities).

 "First, diverse policies provide robustness against changing conditions$\textendash{}$when one policy fails, alternatives with different behavioral characteristics might succeed. Second, diversity is crucial for open-ended learning, where the goal extends beyond solving predefined problems to continually discovering novel capabilities and behaviors".

However, the experiments evaluate only internal QD metrics (QD Score, VS, and qVS), which capture diversity (mostly) and performance within the same environment. Even with experiment 4.4, the authors only examine the best policy. These policies weren't trained to be individually robust, so I'm not sure how meaningful/interesting of a comparison that is compared to looking at population-level robustness (specific suggestions in the Questions section, e.g. percentage of top K policies that achieve a certain reward under parameter shifts). The next point covers weaknesses of this experimental setup.


- Overreliance on internal diversity metrics and lack of qualitative analysis.

The evaluation relies heavily on intrinsic metrics. VS and QD score capture internal notions of diversity but provide limited insight into behavioral / functional variety. qVS multiplies VS by the average return, yielding a crude diversity-quality product that can be misleading: a method with greater qVS may still have lower-quality policies overall. For instance, in Table 1, although AutoQD dominates the Ant environment according to these metrics, the qVS/VS ratio suggests that RegularQD achieves higher average policy performance (i.e., better quality but lower diversity). This raises the question of whether seeing only numerical gains on such metrics truly reflects meaningful improvements.

More broadly, the experiments don't convincingly demonstrate that AutoQD's learned diversity is "useful" (e.g., enhancing robustness or functional adaptability). The only robustness test (section 4.4) examines the best policy under environmental shifts rather than the population-level adaptability that Quality-Diversity methods are designed to provide. Visualizations of the learned descriptor space or representative behaviors would also strengthen the paper by clarifying whether the discovered diversity is interpretable and/or behaviorally meaningful.


[Theorem 1]

- Mismatch between theory and implementation.

Theorem 1 establishes a probabilistic bound under the assumption of independent samples drawn from the occupancy measure. However, the practical estimator in Eq. (6) uses correlated samples collected along trajectories, which violates this assumption. As a result, the theoretical guarantee does not strictly apply to the estimator used in AutoQD. The theorem thus serves primarily as "theoretical motivation" rather than a formal guarantee for the implemented method.

As far as I understand, under mild ergodicity assumptions, Eq. (4) holds in the limit, but the rate of convergence in Theorem 1 would be a lot slower. Part of the reason why this is important is that breaking the i.i.d. assumption is essential to make the method practically usable, it wasn't done just for convenience in the experiments. I would suggest making this clear in the main text of the paper.



- Practical relevance of Theorem 1

There is no guidance on how many trajectories/features are needed for good approximation. The experiments primarily use D = 100, which is likely too small for the theoretical regime where the bound would hold anyway.

**Questions:**

[General]

- The geometry induced by MMD depends on the chosen kernel and might not reflect meaningful behavioral similarity. Do you think other more "structrure-aware" distances (e.g. Wasserstein/Energy) could be used instead and capture the geometry of the space better (or be more interpretable)? (I imagine that the compatibility of MMD and RFFs makes it hard to just switch).

- In the introduction, the authors state: "Crucially, there exists a one-to-one correspondence between policies and their occupancy measures."
This statement is not always true, as you need conditions like full observability and sufficient support over state-action pairs. I recommend rephrasing this to acknowledge the required assumptions (e.g., "under standard assumptions in fully observable MDPs") to avoid overclaiming and be precise about the problem scope.

- In "Future Work", the authors mention integrating AutoQD with gradient-based QD methods, which implicitly suggests that such integration is non-trivial. It would be helpful to clarify why gradient-based QD methods might be harder to use with AutoQD.
One potential challenge is that cwPCA continually updates the affine mapping (A, b) used to compute behavioral descriptors. In a gradient-based setting, this would make the descriptor space a moving target, potentially destabilizing learning.

- Scope clarification: The abstract states that the learned embeddings "are then used as behavioral descriptors for off-the-shelf QD methods," which may overstate generality. The authors should check the wording both in the abstract and conclusion to specify that they focus on CMA-MAE methods and not imply that this is already compatible with the "standard QD algorithms".



[Experiments]

- Visualization of behavior space (what diversity looks like): Can you somehow visualize trajectories to show how policies differ across the learned descriptor space (to better connect the metrics of 4.3 to intuitive diveristy). Additionally, can you visualize what behavioral dimensions cwPCA captures?

- Robustness and usefulness of diversity: Can the authors evaluate the diversity's benefit by testing population-level robustness? For example, measure how many policies in the archive (or the top-K) remain "successful" (or perform well) under dynamics changes (friction, mass). It would also help to report mean or quantile returns under these perturbations in Section 4.4, rather than only the best policy's performance.

- Can you include experiments against gradient-based QD methods?


[Theorem]

- Is there any heuristic to determine when the number of trajectories or random features is sufficient for the MMD embedding to approximate occupancy-measure distances reliably? + In practice, how does QD performance vary with the number of random features or trajectories? (Ablation in Appendix E partially answers the second part of this).


My initial score reflects that I'm unsure whether this paper is ready for ICLR. This is a good piece of work, and I'm keen to increase my score if my concerns are addressed.

---

> ### Author Response · Authors · 2025-11-19
> **Part 1/2**
>
> Thank you for your thorough review of our paper and helpful suggestions. We are glad that you found our work to be original and well-motivated. We have revised the manuscript to address the concerns that you raised and added four more appendices (B, G, I, J) to add further theoretical and experimental support to our arguments. Below, we will answer your questions and concerns and highlight the additions to the paper.
>
> > Do you think other more "structrure-aware" distances could be used instead and capture the geometry of the space better?
>
> A key advantage of MMD is that its induced geometry over occupancy measures can be approximated via L2 distances in a finite-dimensional behavior space using RFFs. Crucially, the uniform convergence property of RFFs enables the probabilistic bound in Theorem 1 for *any* pair of occupancy measures. We are not aware of any analogous result for other distributional distances that would allow finite-dimensional embeddings with similar guarantees. Thus, achieving the same type of probabilistic assurances for approximating their induced geometries in a finite "behavior space" would be challenging. That said, we agree that other distance functions may admit approximations with desirable properties (e.g., interpretability and controllability) and exploring these possibilities is an exciting research direction. We hope AutoQD's approach inspires further work along these lines.
>
> > In the introduction, the authors state [...] I recommend rephrasing this to acknowledge the required assumptions (e.g., "under standard assumptions in fully observable MDPs") to avoid overclaiming and be precise about the problem scope.
>
> You are correct about this. In the revised version, we explicitly acknowledge these assumptions both in introduction and background sections.
>
> > In "Future Work", the authors mention integrating AutoQD with gradient-based QD methods [...]
>
> > Scope clarification: [...]
>
> > Can you include experiments against gradient-based QD methods?
>
> We acknowledge the points regarding the scope of this work and have revised the abstract and conclusion to more accurately reflect our contributions. Throughout the paper, we emphasized that our experiments are focused on CMA-MAE as the base QD algorithm and made sure not to overstate our claims. We would also like to mention some points related to integrating AutoQD with gradient-based methods.
>
> *In principle*, AutoQD provides a general recipe for iteratively learning and refining the behavior space used by a QD algorithm. However, as you pointed out, integrating this iterative refinement with gradient-based QD methods such as PGA-MAP-Elites [1] and PPGA [2] is non-trivial. These methods have more complicated training procedures which involve training critic networks with RL objectives that are known to be sensitive to non-stationary optimization targets. Because AutoQD dynamically updates the behavior space, the associated learning signals for these critics also change over time, which can destabilize training.
>
> As a result, we focused our experiments on CMA-MAE. As a black-box QD method with fewer learned components, it is more robust to the non-stationarities introduced by AutoQD's iterative process. We agree that extending AutoQD to gradient-based QD algorithms is a valuable direction, and we have clarified this point and expanded on its challenges in the conclusion section. A comprehensive exploration gradient-based methods and their integration would require additional methodological analysis, which we leave for future work.
>
> [1] Nilsson, Olle, and Antoine Cully. "Policy gradient assisted map-elites." _Proceedings of the Genetic and Evolutionary Computation Conference_. 2021.
> [2] Batra, Sumeet, et al. "Proximal Policy Gradient Arborescence for Quality Diversity Reinforcement Learning." _The Twelfth International Conference on Learning Representations_.
>
> > Visualization of behavior space [...]
>
> We added Appendix J which provides a qualitative assessment of the behaviors AutoQD discovered. In it, we present trajectories from several policies that display different behaviors across four environments. Notably, we also include examples from the two environments in which AutoQD did not achieve the best performance, and we further discuss the behaviors we observed in those settings. We added these videos to the accompanying code as well, and, upon acceptance, we will open-source our code with the video demonstrations included.
>
> Regarding the visualization of the behavior dimensions captured by cwPCA, these dimensions do not naturally map to specific behavioral variations, making it difficult to evaluate how well they align with human-interpretable concepts. To illustrate behaviors more effectively, we instead adopted a selection approach where we chose a number of policies that spread across the behavior space and selected those that appeared most distinct for our visualizations. This selection process is also described in Appendix J.

---

> ### Author Response · Authors · 2025-11-19
> **Part 2/2**
>
> > Robustness and usefulness of diversity [...]
>
> Based on your suggestion, we added two plots to section 4.4 that demonstrate the number of policies that retain a significant fraction ($p=0.9$ and $p=0.7$, respectively) of the best policy's performance in the original (unaltered) environment, under different variations of the friction coefficient. Intuitively, these plots quantify how many policies from each archive "successfully" adapt to friction changes, in the sense that they maintain near-top-level performance. We also include similar plots for mass changes, the full distribution of returns under variations in both friction and mass, and a more elaborate discussion of these results in Appendix G.
>
> Overall, the results show that under the more strict success criterion ($p=0.9$), AutoQD generally discovers more policies that adapt successfully compared to the baselines. Under the more relaxed criterion ($p=0.7$), AutoQD generally finds the second largest number of successfully adapting policies (behind LSTM-Aurora). AutoQD tends to find more successful policies under more extreme environmental changes (e.g., very high friction), while LSTM-Aurora performs better under milder changes. A more detailed breakdown is provided in Appendix G.
>
> Lastly, we note that our motivation for the adaptation experiments are similar to the damage-recovery scenario in the MAP-Elites paper [1], where unanticipated changes in the environment can cause previously trained policies to fail. In such cases, one would want to select an alternative policy that performs as well as possible in the altered environment. Although having many capable policies is valuable, the ultimate objective is to have *some* policy that performs well, which is why Section 4.4 focused primarily on the performance of the best policy.
>
> [1] Cully, Antoine, et al. "Robots that can adapt like animals." _Nature_ 521.7553 (2015): 503-507.
>
> > Mismatch between theory and implementation. [...]
>
> You are correct in that our algorithm uses $\psi^\pi$ vectors whereas the theoretical analysis considers $\phi^\pi$ as the policy embedding. In the revised version, we added Appendix B which extends our theoretical analysis to bridge this gap and we also discussed this point in Section 3.1. In summary, the fact that $\mathbb{E}[\phi^\pi] = \mathbb{E}[\psi^\pi]$ allows us to transfer the MMD-based guarantees to the $\psi^\pi$ embeddings. Specifically, we show that the distance between the empirical embeddings ($\psi^\pi$) of two policies concentrates around the MMD distance between their occupancy measures, with an additional error term that decays exponentially in $n$. The resulting bound (Eq. 53 in Appendix B) closely parallels the bound in Theorem 1 and retains the same asymptotic behavior.
>
> > Practical relevance of Theorem 1 [...]
>
> > Is there any heuristic to determine when the number of trajectories or random features is sufficient [...]
>
> The primary role of Theorem 1 is to motivate our algorithmic design by showing that our policy embeddings capture meaningful behavioral information as they approximate the geometry of the space of occupancy measures. Because the theorem provides only a probabilistic upper bound, it does not directly guide hyperparameter selection. Although it suggests that increasing $n$ and $D$ yields more accurate embeddings, this accuracy may not directly translate into better downstream QD performance.
>
> Accordingly, we treat the choice of $n$ (number of trajectories) and $D$ (embedding size) as an empirical question that must be answered by considering other factors such as compute and sample budget, as well as experimental results. Our ablations shed light on the impact of these parameters. In Appendix H, we compare different values of $D$, finding that performance is relatively robust, with small gains as we increase the number of features to $D=1000$. In Appendix I (added in the revised version), we also evaluate three choices of $n$ and observe similar patterns. Notably, using as few as $n=2$ trajectories does not substantially degrade performance.
> These findings are also consistent with the second set of experiments added in Appendix I, which compare the stability (i.e., variance) of AutoQD's behavior descriptors with those of the baselines. Across all domains, AutoQD's descriptors are comparatively robust, often exhibiting variance orders of magnitude smaller than that of other methods.
>
> Thank you again for your valuable time and insightful comments. We believe the changes outlined above significantly strengthen the manuscript and have resolved the issues raised. Please let us know if any further clarification or modification is required to satisfy your remaining concerns.

---

> > ### Comment · Reviewer_aLN2 · 2025-11-24
> >
> > The authors have addressed most of my questions and made adequate changes in the latest revision.
> >
> > Below I followup on some points.
> >
> > > On the comparison to gradient-based QD methods
> >
> > It's understandable that the authors don't extend gradient-based QD methods due to the training instabilities already mentioned.
> > However, it remains unclear why there are no comparisons against **gradient-based baselines**.
> >
> > For example, the LSTM-Aurora paper (which performs competitively with AutoQD) compares against PGA-MAP-Elites. As a reviewer, I am unsure whether:
> >
> > 1. including PGA-MAP-Elites/PPGA (or another gradient-based method) is just a lot of work, or
> > 2. these methods significantly outperform AutoQD
> >
> > This is an unresolved concern.
> >
> >
> > > Comments on new appendices
> >
> > The new qualitative behavior analysis in Appendix J is great. It convincingly illustrates the richness of the behaviors discovered by AutoQD.
> >
> > Similarly, the new Figure 4 is very informative. It clearly shows where AutoQD seems to be performing better even in a relatively small setting. I suspect larger-scale experiments would reveal further insights, though that might fall outside the scope of this work.
> >
> >
> > > Rendering of figure
> >
> > There seems to be a rendering issue with Figure 9 in Appendix H. There is some cropped text above "Vendi Score" which seem to suggest there are more plots hidden by some latex issue. Same with figure 10. This may simply be a misaligned figure title, but if additional plots are indeed missing, it would be great to see them.
> >
> >
> >
> > > A further reply on "Mismatch between theory and implementation. ".
> >
> > I'm not sure that $\mathbb{E}[\psi^\pi]=\mathbb{E}[\phi^\pi]$ holds, as stated in the rebuttal.
> > My concern is the following:
> >
> > Step 41 in lemma 1 relies on $t$ going to infinity (infinite horizon occupancy measure). In practive, however, $\psi$ uses a finite horizon $T$ (and its expectation would correspond to a truncated occupancy measure). The difference of $\psi$ and $\phi$ is in the tail mass of the discounted sum. For example, table 5 mentions $\gamma=0.999$ and $T=1000$ which leaves $\gamma^T \approx 36.7$% of geometric probability mass beyond the horizon. Because of the practical necessity of truncation, $\psi$ truncates a part of the discounted sum. Therefore, in practice $\mathbb{E}[\psi^\pi]\neq\mathbb{E}[\phi^\pi]$ because $\psi$ uses a finite sum while $\phi$ uses an infinite one via sampling from the occupancy measure. $\psi$ and $\phi$ having equal expectation when $T\rightarrow\infty$ is not very informative because the whole point of introducing $\psi$ is to make compromises to make the problem tractable (non-i.i.d samples + finite horizon rollout).
> >
> > Also even if the expectations were equal, I'm also unsure whether the bound transfers since MMD depends on the empirical mean not the expectation.
> >
> > The theoretical results still provide useful motivation for the method (and it helps that the method seems to work reasonably well in practice), but I remain unconvinced that the formal bound extends to the actual estimator used in AutoQD (which was my original comment).
> >
> >
> > > Final remarks
> >
> > Overall, the paper has improved considerably. I still believe there is room for further improvement. I am increasing my score to 6.

---

> > > ### Author Response · Authors · 2025-11-25
> > >
> > > We are glad to hear that our additional explanations addressed your main concerns. Below, we respond to the remaining points.
> > >
> > > > On the comparison to gradient-based QD methods
> > >
> > > This is a fair request. We used the same QD algorithm for all QD-based methods so that the only difference comes from how behavior descriptors are obtained. However, your point is well taken, and we are now integrating PGA-ME into our codebase to run an extra baseline with PGA-ME and hand-crafted BDs. (PGA-ME is easier to implement and integrate with our codebase than PPGA.) Once we have the results, we will include them in the paper. Based on prior work comparing PGA-ME with (variants of) CMA-MAE using hand-crafted BDs [1], we expect PGA-ME to perform similarly or slightly better than the current RegularQD baseline. If the experiments are not finished before the rebuttal deadline, we will include them in the camera-ready version. We hope this added baseline addresses your concern.
> > >
> > > [1] Tjanaka, Bryon, et al. "Training diverse high-dimensional controllers by scaling covariance matrix adaptation map-annealing." _IEEE Robotics and Automation Letters_ 8.10 (2023): 6771-6778.
> > >
> > > > Rendering of figure
> > >
> > > Thank you for pointing this out. There was an issue with rendering the titles of these plots. They are fixed in the revised version.
> > >
> > > > A further reply on "Mismatch between theory and implementation."
> > >
> > > You are correct that the argument added in Appendix B assumes an infinite horizon. We added a note to Appendix B (after Lemma 1) to make this clear and to discuss the effect of truncation. Below we summarize the difference.
> > >
> > > If the horizon is finite, that is if $T<\infty$, then $\mathbb{E}[\psi^\pi]$ is not equal to $\mathbb{E}[\phi^\pi]$. However, the difference between the truncated and infinite-horizon $\mathbb{E}[\psi^\pi]$ embeddings is $\mathcal{O}(\gamma^T)$, which corresponds to the tail mass you mentioned. This means that embeddings computed from truncated trajectories have an $\mathcal{O}(\gamma^T)$ bias that should be added to the probabilistic bound. Lowering $\gamma$ or increasing $T$ reduces this bias. In the revised version we compute and account for this additional bias term explicitly. With this addition, the embeddings used in the theoretical result exactly match those used in practice, since both use $\psi$ embeddings and truncated trajectories.
> > >
> > > One explanation for AutoQD's strong empirical performance despite the potentially high bias from $\gamma = 0.999$ is that most behaviors in our domains are cyclical or periodic (hopping, crawling, and similar). As a result, the visited state action pairs are nearly periodic. Truncating early is therefore similar to scaling the embeddings by a constant factor.
> > >
> > > > Also even if the expectations were equal, I am unsure whether the bound transfers since MMD depends on the empirical mean not the expectation.
> > >
> > > We would like to clarify that the proofs in Appendix A and Appendix B bound the distance between *empirical finite-sample embeddings and the true MMD*. Although we use the fact that the expectations are equal (or close in the truncated case), the bound itself concerns embeddings computed as empirical means over $n$ sampled trajectories (hence, the dependence on $n$).
> > >
> > > For instance, the argument in Appendix B, roughly speaking, shows that empirical $\phi$ and $\psi$ embeddings are close to $\mathbb{E}[\phi]$ and $\mathbb{E}[\psi]$, respectively, and that these expectations are equal (or close in the truncated case). Using triangle inequality and union bound, the empirical $\phi$ and $\psi$ vectors must therefore be close. Consequently, the final bound (Eq. 55) shows that the distance between $\psi_P$ and $\psi_Q$ *obtained from $n$ sampled trajectories* is close to the *MMD between $P$ and $Q$*, with high probability.
> > > Appendix A explains how MMD relates to the L2 distance between finite-sample embeddings through three intermediate steps involving $\tilde{\text{MMD}}$ and $\hat{\text{MMD}}$. We also revised Appendix A slightly to clarify the use of concentration bounds.
> > >
> > > Thank you once again for your thoughtful suggestions. We hope the explanations above have resolved the remaining concerns, and we'd be happy to clarify anything further if needed.

---

### Official Review · Reviewer_Xeky · 2025-11-03

**Soundness:** 3
**Presentation:** 3
**Contribution:** 3
**Rating:** 6
**Confidence:** 3

**Summary:**

The paper introduces AutoQD, a method that automatically discovers behavioral descriptors for Quality-Diversity optimization. It proposes embedding a policy's unique state-action occupancy measure using Random Fourier Features, such that the L2 distance between embeddings approximates the Maximum Mean Discrepancy (MMD). A Calibrated Weighted PCA is then used to project these embeddings into a low-dimensional descriptor space, prioritizing variations among high-performing policies. This enables standard QD algorithms like CMA-MAE to discover diverse and effective behaviors without requiring hand-crafted, domain-specific features.

**Strengths:**

- The paper is theoretically sound, rigorously connecting occupancy measures, MMD, and Random Fourier Features to create a principled and efficient metric for behavioral distance.


- The proposed policy embedding method is a versatile contribution with significant potential beyond QD. This technique for representing policy behavior could be applied to other RL tasks, making it a valuable tool for the broader community.


- The experiments are extensive and includes a diverse set of environments even thought they are all low dimensional in the state space observations and clearly show that AutoQD is able to outperform sota algorithms in terms of GT QD Score.

**Weaknesses:**

- There exist a gap between the theoretical guarantees and the practical implementation of the policy embedding. Theorem 1 provides a powerful result for embeddings ($\phi^\pi$) constructed from i.i.d. samples drawn from the occupancy measure. However, the paper acknowledges that this sampling strategy is too inefficient for practical use. Instead, the algorithm uses a different estimator ($ \psi^\pi$ from Eq. 6) that averages features over all transitions in a trajectory. The paper lacks a formal analysis of how this practical estimator affects the approximation quality, leaving a gap in the theoretical justification of the implemented algorithm.

- The empirical validation, while strong, is exclusively focused on environments with low-dimensional, vector-based state spaces. The scalability of the RFF-based embedding to high-dimensional observations, such as images, remains unproven. Such environments could pose significant challenges, as the curse of dimensionality might render the MMD approximation less effective or require an impractically large embedding dimension (D).

- While the proposed method is interesting, it seems to completely disregard temporal information about the order of events. Two policies that produce the same state-action occupancy distribution but follow a different temporal order could lead to very different decision-making behaviors. Therefore, they should be accurately identified as distinct (or diverse) policies.

**Questions:**

See the Weaknesses points.

---

> ### Author Response · Authors · 2025-11-19
>
> We are thankful for your time and help in reviewing our paper. We are glad to hear that you found our work to be theoretically sound and with significant potential beyond QD. We have revised the manuscript based on your feedback. Below, we address the three concerns you raised regarding its weaknesses.
>
> > There exist a gap between the theoretical guarantees and the practical implementation of the policy embedding.
>
> You are correct in that our algorithm uses $\psi^\pi$ vectors whereas the theoretical analysis considers $\phi^\pi$ as the policy embedding. In the revised version, we added Appendix B which extends our theoretical analysis to bridge this gap and we also discussed this point in Section 3.1. In summary, the fact that $\mathbb{E}[\phi^\pi] = \mathbb{E}[\psi^\pi]$ allows us to transfer the MMD-based guarantees to the $\psi^\pi$ embeddings. Specifically, we show that the distance between the empirical embeddings ($\psi^\pi$) of two policies concentrates around the MMD distance between their occupancy measures, with an additional error term that also decays exponentially in $n$. The resulting bound (Eq. 53 in Appendix B) closely parallels the bound in Theorem 1 and retains the same asymptotic behavior.
>
> > The scalability of the RFF-based embedding to high-dimensional observations, such as images, remains unproven.
>
> This criticism is fair and we have listed this as a potential avenue for future work. Our theoretical results suggest that the size of the RFF embeddings ($D$) should increase at least linearly with the size of the observation vectors ($d$) to maintain a fixed approximation error. Therefore, in high-dimensional, image-based observation spaces we may require very large RFF embeddings which could introduce additional challenges. A practical strategy in such settings would be to first encode images using a pre-trained or online-trained vision model and apply RFFs in the resulting latent space. While a full investigation is beyond the scope of this paper, we view AutoQD as a first step upon which such extensions can be built.
>
> > the proposed method [...] seems to completely disregard temporal information about the order of events. Two policies that produce the same state-action occupancy distribution but follow a different temporal order could lead to very different decision-making behaviors.
>
> We would like to clarify that this is not the case. Under standard RL assumptions in fully observable MDPs, occupancy measures are equivalent to Markovian (memoryless) policies. Therefore, if two policies produce the same state-action occupancy distribution, they must be equal. Indeed, our cited reference in Section 2.1 ([1]) shows how one can recover the unique policy that induces a given occupancy distribution. Intuitively, the discount factor $\gamma < 1$ preserves temporal information: a state visited later contributes less to the discounted occupancy measure than a state visited earlier, so two policies with different visitation orders cannot produce the same distribution.
>
> [1] Puterman, Martin L. _Markov decision processes: discrete stochastic dynamic programming_. John Wiley & Sons, 2014.
>
>
>
> We hope that these explanations and revisions sufficiently address your concerns. Please let us know if any further clarification or modification is required to satisfy your remaining concerns.

---

### Author Response · Authors · 2025-12-03
**Summary of Rebuttal Discussions and Revisions**

Dear Area Chair,

As the discussion period is coming to an end, we are writing to summarize the productive exchange we had with the reviewers and the improvements we have made to the manuscript.

The reviewers had positive sentiment (giving scores of 6 and 4) and they raised some good points that we addressed in the rebuttal. Even the reviewers who gave a 4 had a generally positive view of the paper, with Reviewer aLN2 acknowledging explicitly that **"This is a good piece of work, and I'm keen to increase my score if my concerns are addressed."** We believe that our rebuttal has answered the main points raised by the reviewers, as evident by Reviewer aLN2's final response.

We provided comprehensive answers to the points raised by the reviewers, resulting in four new Appendices (B, G, I, J) and a revision of the main text. Below, we summarize the key outcomes of the discussion and the resulting improvements.

1. **Bridging Theory and Practice (Appendix B)**
   A primary concern shared by Reviewers Xeky, aLN2, and sHcp was the perceived gap between our theoretical bounds (based on infinite-horizon embeddings $\phi^\pi$) and our practical implementation (using finite-horizon estimators $\psi^\pi$). We addressed this by adding Appendix B, which provides a formal derivation linking the two. We extended our theoretical result to show that the distance between empirical $\psi^\pi$ embeddings concentrates around the true MMD, with two additional error terms (with similar asymptotic behaviors) resulting from using $\psi$ vectors instead of $\phi$ vectors and truncating the episodes after $T$ steps. This addition fully aligns the theory with the implementation. We are particularly grateful to Reviewer aLN2 for the discussion that led to the extended theoretical derivation.

2. **Qualitative Analysis & Behavior Visualization (Appendix J)**
   To address Reviewers cGSg and aLN2's request for better visualization of the learned diversity, we added Appendix J and supplementary videos. We visualized trajectories from policies across the behavior space, demonstrating distinct behavioral modes. Reviewer aLN2 praised this addition, stating: "The new qualitative behavior analysis in Appendix J is great. It convincingly illustrates the richness of the behaviors discovered by AutoQD."

3. **Demonstrating Robustness & Usefulness (Appendix G & Section 4.4)**
   Reviewer aLN2 asked for more comprehensive evidence that the diversity discovered by AutoQD is practically useful beyond internal metrics. In response, we expanded Section 4.4 and added Appendix G, analyzing population-level robustness. We measured how many policies maintain high performance under significant physical perturbations (friction and mass changes). The results demonstrate that AutoQD is capable of discovering a large number of policies capable of adapting to environmental shifts.

4. **Stability & Sample Efficiency (Appendix I)**
   Addressing concerns from Reviewers cGSg, sHcp, and aLN2 regarding sample efficiency and stability, we added Appendix I. We performed ablations on the number of trajectories ($n$) used for embeddings, showing that AutoQD maintains strong performance even with as few as $n=2$ trajectories. Furthermore, we analyzed the variance of the generated behavior descriptors, showing that AutoQD's descriptors exhibit significantly higher stability (lower variance) compared to baseline methods.

5. **Clarifying Scope & Gradient-Based Comparisons**
   We clarified the scope of our contribution in the abstract and conclusion, emphasizing our focus on CMA-MAE as the base QD optimizer. In the paper, we explicitly stated that replacing CMA-MAE with gradient-based QD methods can improve the performance, but adapting AutoQD's policy embedding strategy to gradient-based methods is non-trivial and left for future work. In response to Reviewer aLN2's suggestion, we also committed to including an experiment that uses hand-crafted BDs and a gradient-based QD method (PGA-MAP-Elites) in the final version to further contextualize AutoQD's performance gains with the benefits attainable by using more complex base QD algorithm.

We believe AutoQD offers a principled, theoretically grounded solution to the hand-crafted descriptor bottleneck in Quality-Diversity optimization. Given the theoretical and empirical strengthening in the revision and the explicit confirmation from Reviewer aLN2, we hope that our extensive rebuttal provides a strong basis for acceptance.

Thank you for your time and consideration.
Sincerely,
The Authors

---

### Meta-Review · Area_Chair_gkjA · 2026-01-03

**Summary:**

This paper introduces the AutoQD method that solves one of the most important issues in quality diversity optimization algorithms, namely their reliance on hand-crafted Behavioral Descriptors. The AutoQD method uses Random Fourier Features to approximate the MMD between occupancy measures for policy embeddings and combines with QD algorithms to alternate between policy optimization and Behavioral Diversity steps. The authors also provide theoretical results showing the learned embeddings converge to the true MMD as the number of samples and embedding dimensions increase. Some of the main concerns raised by reviewers were about the number of trajectories that are needed to compute the Behavioral Embeddings, and how many such trajectories are necessary to achieve this, and the mismatch between theory and implementation. These concerns were addressed by the authors, causing some of the low scoring reviewers to signal their willingness to update their score. Thus I recommend acceptance of this work.

**Reviewer Concerns:**

Reviewers’ main concerns centered on the sample complexity required to compute the Behavioral Embeddings specifically, how many trajectories are needed in practice and on an apparent mismatch between the theory and the implementation. The authors addressed these issues, and several of the lower-scoring reviewers indicated they would be willing to update their scores accordingly.

**Reviewer Scores:**

Some of the lowest scoring reviewers raised their scores.

---

### Decision · Program_Chairs · 2026-01-26

Accept (Poster)